# Cortical reactivation of spatial and non-spatial features coordinates with hippocampus to form a memory dialogue

HaoRan Chang [1] ✉, Ingrid M. Esteves[1], Adam R. Neumann[1], Majid H. Mohajerani[1,2] & Bruce L. McNaughton[1,3]

Episodic memories comprise diverse attributes of experience distributed across neocortical areas. The hippocampus is integral to rapidly binding these diffuse representations, as they occur, to be later reinstated. However, the nature of the information exchanged during this hippocampal-cortical dialogue remains poorly understood. A recent study has shown that the secondary motor cortex carries two types of representations: place cell-like activity, which were impaired by hippocampal lesions, and responses tied to visuo-tactile cues, which became more pronounced following hippocampal lesions. Using two-photon $Ca^{2+}$ imaging to record neuronal activities in the secondary motor cortex of male Thy1-GCaMP6s mice, we assessed the cortical retrieval of spatial and non-spatial attributes from previous explorations in a virtual environment. We show that, following navigation, spontaneous resting state reactivations convey varying degrees of spatial (trajectory sequences) and non-spatial (visuo-tactile attributes) information, while reactivations of non-spatial attributes tend to precede reactivations of spatial representations surrounding hippocampal sharp-wave ripples.

The encoding, storage and retrieval of episodic memories require a carefully orchestrated exchange of information between the hippo-campus and the neocortex[1–4]. During periods of active behaviour, unique experiences are comprised of diverse attributes that span a multitude of modalities and cortical sites. The task of associating and storing these distributed patterns of activity as they occur, so that they may be recalled at later times, has been attributed to the hippocampal network[1,2,4]. An important line of experimental evidence reinforcing this theoretical principle stems from the study of reactivation; during offline periods (i.e., sleep and quiet-wakeful states), patterns of activity that are related to previous experiences are spontaneously and repeatedly reinstated. Consistent with the notion of a distributed memory system, multiple cortical regions have been implicated in the reactivation of behavioural features, which are congruous with the

known functions of their respective regions (e.g., refs. 5–10). For instance, the retrosplenial cortex, a region responsible for the encoding of stable landmarks, reactivates for landmark locations[11], while the medial prefrontal cortex, implicated in cognitive flexibility and rule learning, reactivates for task rules[12]. The reactivations of these distributed features are also temporally coordinated across cortical regions[13].

In rodents, a substantial portion of the variability in cellular activity of hippocampal neurons is explained by space, whereby individual neurons' discharges are correlated to specific locations in a spatial environment—these are known as place cells[14]. Accordingly, reactivated patterns in the medial temporal lobe and medial prefrontal cortex manifest themselves as sequences of trajectories undertaken in a previously explored environment[15–19]. As is the case between cortical

[1]Canadian Centre for Behavioural Neuroscience, Department of Neuroscience, University of Lethbridge, 4401 University Drive, Lethbridge T1K 3M4 AB, Canada. [2]Department of Psychiatry, Douglas Hospital Research Centre, McGill University, 6875 Boulevard LaSalle, Verdun, QC H4H 1R3, Canada. [3]Department of Neurobiology and Behavior, University of California, 2205 McGaugh Hall, Irvine 92697 CA, USA. ✉e-mail: haoran.chang@mail.mcgill.ca

regions, the hippocampus reactivates in coordination with many cortical areas[5–7,9,20]. These coordinated reactivation events often occur in conjunction with hippocampal sharp-wave ripples (SWRs)—discrete high frequency events expressed by synchronous CA1 neuronal populations during offline periods—which are temporally coupled with transient cortical activities[21–28]. Though a continuous gradient likely marks the timing between hippocampal and neocortical activations[29], the onset of cortical reactivations typically precedes that of hippocampal SWRs by an order of ~50–200 ms[5,8,9,12]. Conversely, hippocampal population activities reliably trigger responses in the cortex[6,7,21,24,27], while the content of hippocampal activities during SWRs can predict subsequent cortical patterns[9].

In principle, if the hippocampus were to provide a set of associative links − an index − pointing to specialized patterns distributed over cortical regions[2,30], then it follows that the contents of the activities expressed during coordinated reactivations between the hippocampus and the neocortex should share similar features from previous experiences. Such was the case reported by multiple studies, which collectively mapped several cortical areas that reactivated for patterns complementary to those reactivated by the hippocampus[5–7,20]. However, in these studies, the shared features in question were correlated over the spatial dimension, where the contents of reactivations were linked to specific locations from previous explorations. In one exceptional study, the contents of the reactivated patterns observed in the auditory cortex were related to specific tonal stimuli[9]. However, the

exact functional relevance of the patterns reactivated in conjunction by the hippocampus could not be determined. Given that diverse cortical areas encompass a wide range of cognitive processes, the functional links that would permit associations to be formed between the hippocampus and the cortex remains to be elucidated. Specifically, the spatial and non-spatial aspects of experiences, which seemingly constitute an important functional basis for the hippocampal-cortical dialogue[3] in a distributed memory system, require further reconciliation.

Recently, a study has reported that, during a virtual spatial navigation task, two kinds of representations were concurrently supported by primary and secondary neocortical regions: place cell-like activities which were impaired by hippocampal lesion, and responses related to visuo-tactile cues which became more pronounced following hippocampal lesion[31]. Premised on this finding, the current study aimed to explore whether these distinct representations, which are likely of hippocampal and cortical origin respectively, are reactivated concurrently by the same cortical region, and if so, whether they engage in interactions that are reflective of an exchange of information between the hippocampus and the neocortex. Using two-photon calcium imaging, we simultaneously recorded populations of neurons in the superficial layers (LII/III) of the secondary motor cortex (M2) of Thy1-GCaMP6s transgenic mice ($n = 14$ animals; Fig. 1b; Supplementary Table 1; ~19 frames-per-second). Water-restricted mice

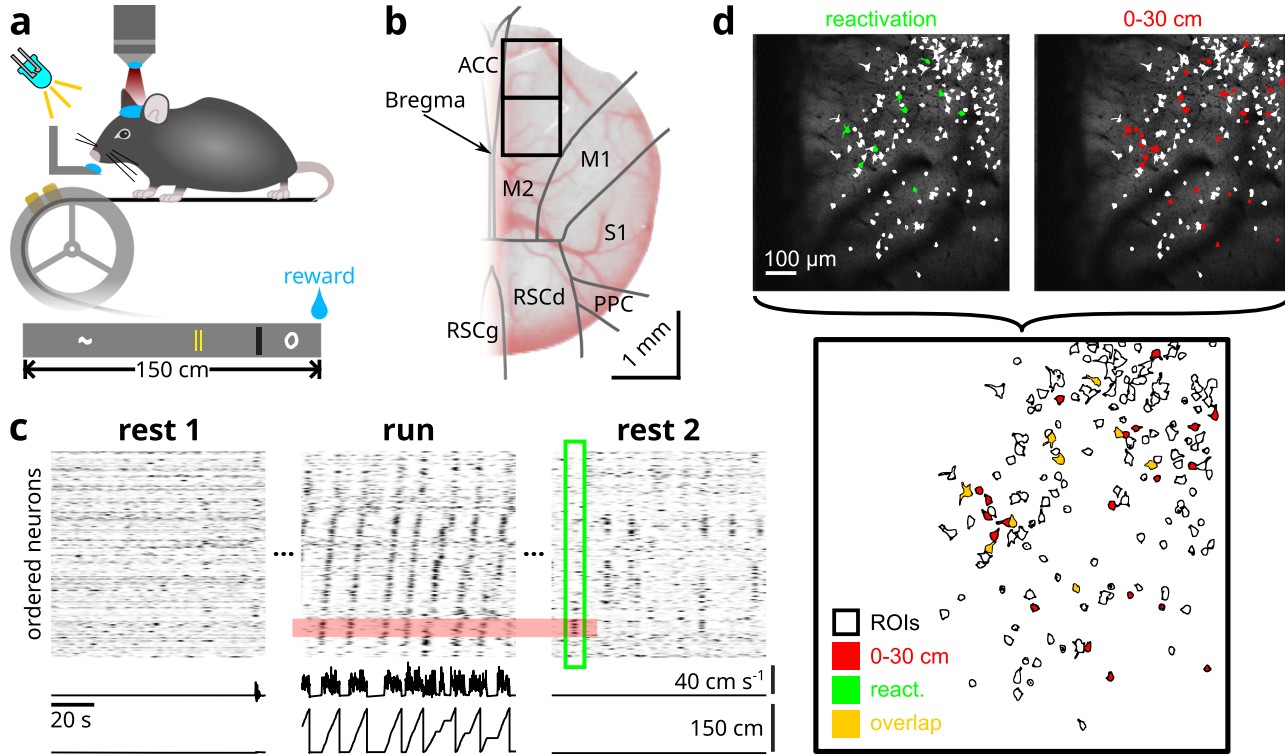

**Fig. 1 | Two-photon imaging and behavioural paradigm. a** Water-deprived mice were head-restrained over a 150 cm long treadmill belt, over which were mounted several visuo-tactile cues. An LED light illuminated the path in front of the animals so that they are able to see the incoming cues. **b** Example of a cranial window implant. Black boxes delineate the two imaging windows, of which one is chosen for each mouse on the basis of signal quality and the presence of bone-regrowth. Regional boundaries were determined from the Allen Common Coordinate Framework atlas (CCF v3). **c** Each imaging session was divided into three 10–20 min blocks. Before (REST1) and after (REST2) exploration (RUN), animals rested quietly over the belt, during which the treadmill was clamped. A 100 s segment from each imaging block is illustrated for one example session (out of $n = 86$ recording

sessions in total). The time courses of deconvolved $\Delta F/F_0$ for all simultaneously imaged neurons were sorted by the location of peak activity during RUN. Animal position and linear velocity are shown below. Notice that, during REST2, co-active ensembles of neurons are clustered with respect to the sequential activity patterns over spatial locations during RUN. **d** ROIs of neurons that were active during the 1 s time window surrounding the single reactivation event delineated in **c** (green), and those of neurons that were active at locations 0–30 cm, roughly corresponding to the span of the trajectory encoded by the ensemble of reactivating neurons (cf. Fig. 3a). The overlapping ROIs between those two groups are shown. All neurons that were part of the reactivation event were also active in the first 30 cm during behaviour.

were trained to navigate on a linear treadmill for a drop of sucrose water at the end of each lap (Fig. 1a). The treadmill consists of a 150 cm long belt lined with four distinct visuo-tactile cues. To characterize awake reactivation patterns, we acquired resting state activity for 10–20 min before (REST1) and after (REST2) virtual exploration (RUN) (Fig. 1c).

We found that, in the resting period following virtual navigation, awake reactivation events carry information that is, in varying degrees, related to trajectory segments in space or visuo-tactile attributes. Reactivations that are biased towards visuo-tactile features tend to occur earlier in time relative to hippocampal SWRs, whereas reactivations that are more related to spatial trajectories tend to occur later. Furthermore, concurrent reactivations of cue and trajectory information reinstate similar features from previous experiences, where the reactivated trajectory segments tend to coincide with the locations of the reactivated cues. These results are commensurate with the theorized notion that cortical reactivation of non-spatial attributes may act as partial information to seed the hippocampal retrieval of associated spatial sequences. These spatial representations are then propagated back to the cortex, hence forming a functional cortical-hippocampal-cortical loop (cf. ref. 9).

## Results

### M2 ensemble dynamics during resting state

We began with a characterization of the resting state activity in the secondary motor cortex, while screening for evidence of reactivation of task-related patterns. At the population level, neuronal activities exhibited higher synchrony in the ultra-slow frequency range (0.05–0.5 Hz) during RUN, as compared to during awake quiescent periods (Supplementary Fig. 1a, b, d, g). This synchrony likely arose from the entrainment of cortical activities by locomotion (Supplementary Fig. 1c). In parallel, a decrease in the rate of calcium transients was also associated with running (Supplementary Fig. 1f), a phenomenon that has been previously described[32]. In contrast, during awake quiescence, population activities expressed stronger power in the slow oscillatory ranges (1–10 Hz) (Supplementary Fig. 1b, e), in accordance with a synchronized cortical state (see refs. 33,34). Overall, the secondary motor cortex exhibited distinct population dynamics across awake behavioural states.

Consistent with previous reports[11,19,35], activities during the resting periods were characterized by spontaneously co-activating groups of neurons (Fig. 1c; Fig. 2a) − henceforth referred to as ensembles. Between REST1 and REST2, ensembles were composed of a

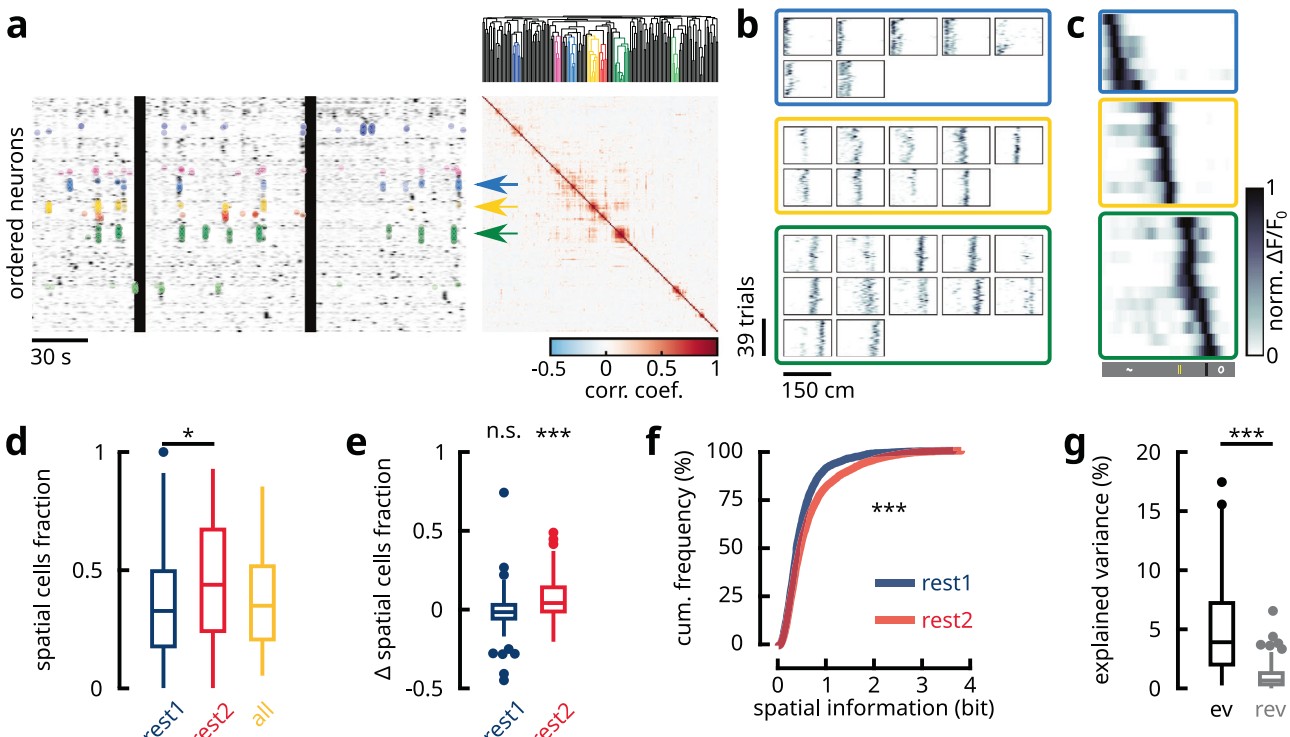

**Fig. 2 | M2 resting ensembles preferentially recruit neurons expressing high spatial information. a** Illustration of hierarchical clustering method used to detect synchronous neuronal ensembles during rest. A 1 min segment of REST2 from the same recording as Fig. 1c–d is shown. Neurons were sorted by similarity in time course vectors. Black vertical bands highlight movement epochs, which were excluded from analysis. The Pearson correlation matrix between the time-series of neuron pairs is illustrated, along with the dendrogram obtained from hierarchical clustering. Neurons part of a synchronous ensemble were grouped by the same colour. Coloured dots over the time-series mark reactivation events detected for individual ensembles. **b** The mean deconvolved $\Delta F/F_0$ as a function of position and laps for all individual neurons in the three synchronous ensembles in **a**. Neurons were sorted by place field location. Notice that neurons of the same ensemble share neighbouring place fields. **c** The mean activity of the neurons in **b** as a function of position. **d** The fraction of spatially-selective neurons that were part of REST1 and REST2 synchronous ensembles, and the overall fraction of spatial cells in all recording sessions ($n = 14$ mice; $n = 86$ sessions; $p = 0.034$ for REST1-REST2; not

significant between other groups; Kruskal-Wallis H test). **e** Difference between the fraction of spatially-selective neurons within synchronous ensembles and the overall fraction of spatial cells ($n = 14$ mice; $n = 86$ sessions). REST2 ensembles contained a significantly higher fraction of spatial cells ($p = 7.299 \times 10^{-5}$; one-tailed Wilcoxon signed rank test for the null hypothesis of median smaller or equal to zero), while REST1 ensembles showed no significant deviation from baseline ($p = 0.92$). **f** Cumulative distribution functions of spatial information for REST1 and REST2 neurons that were part of synchronous ensembles ($n = 2672$ REST1 and $n = 3613$ REST2 neurons; $p = 1.773 \times 10^{-15}$ two-tailed Kolmogorov-Smirnov test; $p = 1.22 \times 10^{-18}$ two-tailed Mann-Whitney U-test). **g** Percentage of explained variance (ev) and reverse explained variance (rev) in all imaging sessions ($n = 14$ mice; $n = 86$ sessions; $p = 1.815 \times 10^{-15}$; paired-sample one-tailed Wilcoxon signed rank test for the null hypothesis of median smaller or equal to zero). All box plots show the median (line), the first and last quartiles (box), the minimum and maximum values (whiskers) and the outliers (circles). Source data are provided as a Source Data file.

comparable number of neurons (Supplementary Fig. 2a, b) with an average of 6.92 and 7.75 members respectively. However, a greater number of ensembles were detected in REST2 (Supplementary Fig. 2c), which suggests that the network, following locomotion, exhibited more stereotyped patterns of activity. We tested the latter hypothesis by carrying out a principal component analysis on the correlation matrices of time series vectors for REST1 and REST2 (Supplementary Fig. 2e). The results show that fewer components were needed to explain a greater fraction of the total variance in the neuronal population of REST2, which affirms the notion that the population activity contained more recurring patterns. Moreover, REST2 ensembles activated at a lower rate than REST1 ensembles (Supplementary Fig. 2d). Taken together, these results suggest that following exposure to a virtual environment, the spontaneous activities of the secondary motor network became more stereotyped, with similar patterns of population activity being reinstated over time.

To understand how these synchronous activities could be related to functional patterns, we examined the characteristics of the ensemble neurons during active behaviour. In REST2, synchronous ensembles preferentially recruited neurons that expressed spatial-selectivity during active locomotion. Indeed, REST2 ensembles contained a median of 43.8% spatially-selective cells, higher than the 32.6% in REST1 ensembles (Fig. 2d). When the difference is taken between these percentages and the total fraction of spatially-selective cells in a given session, we found that the median of this difference was not significantly different from zero for REST1, while it was higher than zero for REST2 (Fig. 2e). This suggests that the constituent neurons of REST2 ensembles were more likely to include cells that were spatially selective during RUN than REST1 ensembles. A more robust approach for testing this relationship is through a hypergeometric test, which aims to model the probability that a certain fraction is obtained by drawing at chance from a population. This test revealed that a significantly higher proportion of REST2 ensembles had a high composition of spatially-selective cells (Supplementary Fig. 3). Overall, REST2 ensemble neurons expressed higher spatial information content (Fig. 2f).

This propensity for REST2 ensembles to recruit neurons with high spatial information suggest that M2 may be reactivating features related to recent experiences. The first evidence for reactivation came from explained variance analysis[23], which revealed that REST2 population activity accounted for a higher percentage of the variance in RUN than REST1 activities (Fig. 2g). Given that activity patterns during active behaviour were characterized by continuous sequences of place cell-like activity, this result suggested that most of the variance exists in the correlational structure between neurons with nearby place fields. Indeed, REST2 ensembles tend to contain spatial cells with place fields that were in proximity to each other (Fig. 1c, d; Fig. 2b, c). Next, we investigated the significance of this organization.

## Cue and trajectory information are jointly reactivated

Our previous findings[31] suggested that the secondary motor cortex supports two distinct encodings during virtual navigation: place cell-like activities which were impaired by hippocampal lesion, and responses associated with visuo-tactile cues which became more pronounced following hippocampal lesion (Supplementary Fig. 4). We investigated whether these separate neuronal representations may be preferentially reactivated in cortex. Intriguingly, resting-state ensembles that consisted of spatially-selective cells sharing neighbouring place fields could be classified into two separate categories based on the contents of their reactivated features. On the one hand, we noted ensembles that were composed of neurons with nearly identical spatial tuning profiles. These cells often supported multiple place fields (Fig. 3a, b) that strongly overlapped with the locations of visuo-tactile cues (Fig. 3f), while the same place fields tended to be shared between all ensemble members (Fig. 3a, d). Given their propensity to encode

cue information, we dubbed these groups cue ensembles. On the other hand, we noticed ensembles that were composed of members that shared place fields in proximity to one another (Fig. 2b, c; Fig. 3a; Supplementary Fig. 5), although they were not completely overlapping (Fig. 3d). Collectively, they formed short segments of continuous trajectories in space with a median length of 36 cm for REST1 and 33 cm for REST2, corresponding to 24% and 22% of the length of the environment respectively (Fig. 3a; Supplementary Fig. 6). These ensembles will be termed trajectory ensembles.

Both trajectory and cue ensembles contained a significantly higher number of spatially-selective cells than the ensembles that were unclassified (Fig. 3e). This suggests that the reactivated contents supported by these groups were highly specific to spatial features encoded during locomotion. In contrast, the width of the place fields in cue ensemble neurons was markedly narrower than that of trajectory ensembles (Fig. 3c). Indeed, the median of the average width of place fields within cue ensembles was 35.4 cm, which was lower than the 45.6 cm in trajectory ensemble neurons (note that place field widths and trajectory lengths were quantified using different methods, which makes it appear as though place fields are wider than the trajectories that they form; see Methods). This indicates that cue ensemble members have inherently sharper tuning profiles, which likely reflects a primary/secondary response towards cue stimuli. Overall, trajectory ensembles were more than three times as prevalent as cue ensembles in both REST1 and REST2, although a greater fraction of REST2 ensembles could be placed into either of these two categories (Fig. 3g). These fractions are consistent with the comparatively low proportions of cue-responding neurons within the total population of spatially-selective cells, which was estimated to be 16.51% (Supplementary Fig. 7). Furthermore, a comparable number of neurons comprised the two ensemble classes (Supplementary Fig. 10c), signifying that the classification of these ensembles was not biased by sample size. During locomotion, neurons belonging to cue ensembles showed stronger activation than trajectory ensemble neurons (Supplementary Fig. 8a). This trend was reversed, however, during resting states (Supplementary Fig. 8c). In parallel, the mean rates of calcium transients conveyed by trajectory ensemble neurons was slightly higher compared to cue ensemble neurons during rest, while comparable rates were reported during locomotion (Supplementary Fig. 8b, d). This was corroborated by a heightened rate of reactivations during REST2 for trajectory ensembles compared to cue ensembles (Supplementary Fig. 8e, f). Altogether, these results suggest that the reactivation of two separate types of neural representations could be observed in the secondary motor cortex.

Previous studies showed that, during replay of previous behaviour-correlated patterns, the activity sequences undergo temporal compression[17,18]. In relation to our current findings, an interesting hypothesis may be proposed. On the one hand, if cue ensembles contain neurons that, during behaviour, respond simultaneously to the sensing of cues, then, owing to the lack of a sequential structure, they should not undergo temporal compression during reactivation. On the other hand, trajectory sequences are expected to undergo compression as previously reported. Such was indeed the case; we reported median values of optimal compression factors of 2× for cue ensembles, compared to 30× for trajectory ensembles (Supplementary Fig. 9). Note that, owing to the low temporal sampling rate, these estimated values are expected to fall over a broad confidence intervals range (cf. Supplementary Methods).

In spite of the tendencies for ensembles to encode cue versus trajectory information, there remains the possibility for ensembles to be composed of a mixture of cue-responsive and place-responsive cells, which themselves may be encoding for conjunctive features. To test this possibility, two models were devised and fitted to the time-series of each individual ensemble neuron. The first model assumes that the neuron's response tuning curve abides by a Gaussian function

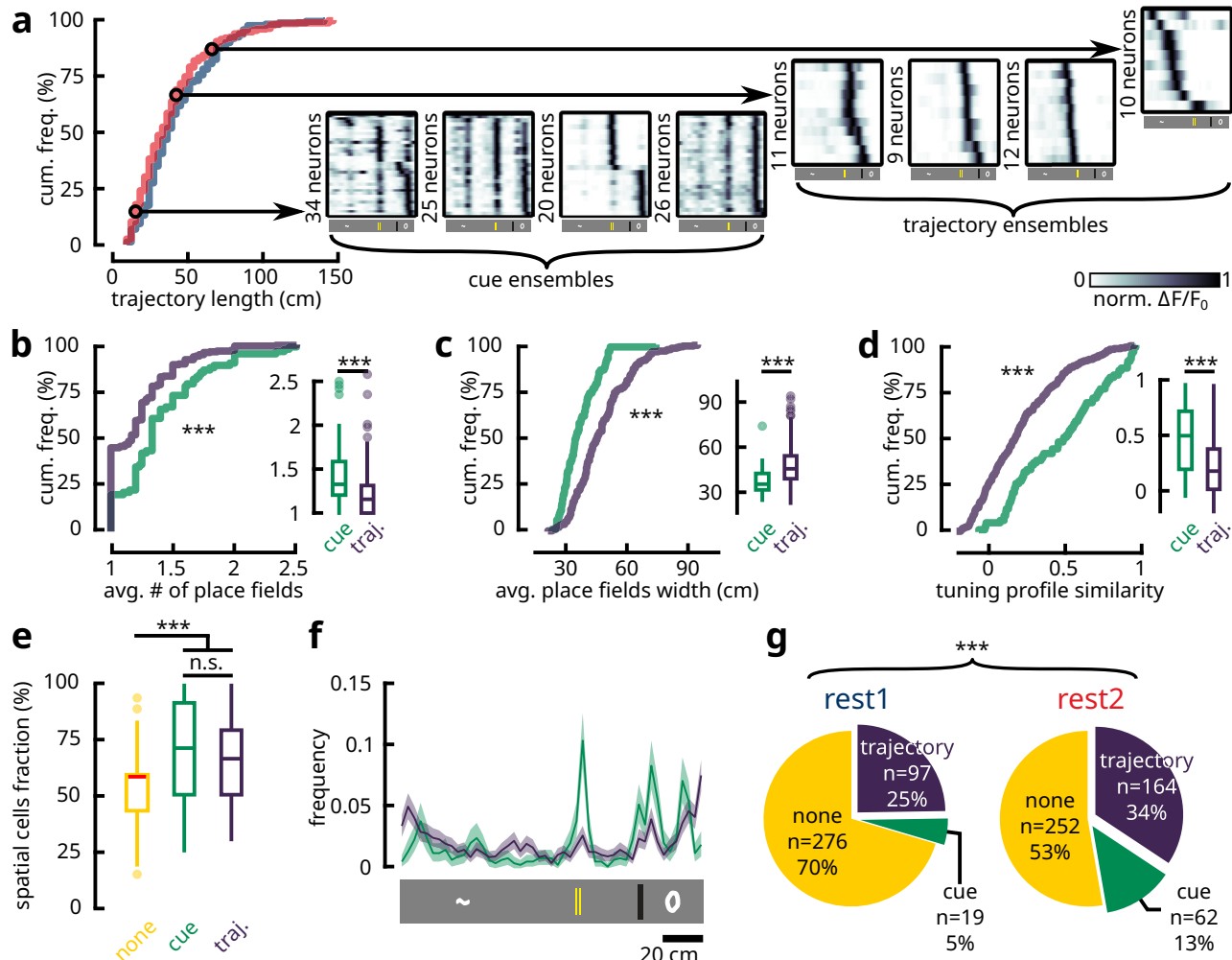

**Fig. 3 | Resting state ensembles reactivate spatial and non-spatial features.**
**a** Cumulative distribution functions of the length of trajectories encoded by REST1 and REST2 ensembles ($n = 160$ REST1 trajectories; $n = 325$ REST2 trajectories; $p = 0.059$; two-tailed Kolmogorov-Smirnov test). A subset of these trajectories with a length shorter than 30 cm were composed of neurons with nearly identical tuning profiles, coinciding with the locations of visuo-tactile cues. Four examples of cue ensembles and trajectory ensembles are shown, with the mean activity of ensemble neurons as a function of position, sorted by place field location. Different metrics used to compare the spatial contents reactivated by cue and trajectory ensembles (two-sample two-tailed Kolmogorov-Smirnov test; $p = 2.932 \times 10^{-5}$, $8.102 \times 10^{-8}$, $7.786 \times 10^{-8}$ from **b** to **d**; two-tailed Mann-Whitney U-test; $p = 1.45 \times 10^{-6}$, $1.769 \times 10^{-11}$, $4.025 \times 10^{-11}$ from **b** to **d**). For each ensemble ($n = 81$ cue ensembles; $n = 261$ trajectory ensembles; $n = 528$ unclassified ensembles), neurons that were identified as spatially-selective cells were extracted. The average number of place fields (**b**) and the average width of the place fields (**c**) were computed for each ensemble. Additionally, the Pearson correlation matrix of the spatial tuning profiles between spatial cell pairs was obtained, and the average value in the upper triangle of the correlation matrix was taken (**d**). **e** The

percentage of spatially-selective neurons in cue, trajectory and unclassified ensembles. Both cue and trajectory ensembles carried more spatially-selective cells than unclassified ensembles, but the fractions were comparable between the two categories of interest (Kruskal-Wallis One-way ANOVA $p = 1.787 \times 10^{-7}$; post-hoc tests report Bonferroni-adjusted p-values: none-cue $p = 2.607 \times 10^{-6}$, none-traj $p = 6.857 \times 10^{-7}$, cue-traj $p = 1$). For fair comparisons, unclassified ensembles with less than 3 spatially-selective cells were omitted ($n = 72$ remaining).
**f** Histogramic distributions of place field centres ($n = 741$ place fields for cue ensembles; $n = 1676$ place fields for trajectory ensembles) for spatially-selective cells that were recruited by cue and trajectory ensembles. Shaded area denotes 95% bootstrapped confidence interval. **g** Pie charts for the portion of trajectory, cue and unclassified (none) ensembles in REST1 and REST2. The proportions of cue and trajectory ensembles were significantly increased from REST1 to REST2 (two-tailed $\chi^2$ test; $p = 7.041 \times 10^{-8}$; $\chi^2 = 32.94$; df = 2). All box plots show the median (line), the first and last quartiles (box), the minimum and maximum values (whiskers) and the outliers (circles). Source data are provided as a Source Data file.

over spatial locations, acting as a first-order approximation of a place cell. The second model fits a distinct firing rate at each cue location with a constant baseline firing rate, therefore reflecting a sensory stimulus-driven neuronal response. The distribution of ratios between the goodness-of-fit of the two models (measured as the likelihood ratio) did not express bimodality, which confirms the hypothesis that ensembles or ensemble neurons encode for conjunctive features between cue and trajectory (Supplementary Fig. 10a). However, neurons that belonged under the class labels of cue or trajectory ensembles did show a significant bias over these ratios (Supplementary

Fig. 10a, b), which confirms that ensembles have distinct propensities to encode for cue and trajectory. This is further corroborated by the fact that cue ensembles were more accurate at decoding individual cue identities than trajectory ensembles using a Bayesian paradigm (Supplementary Fig. 10d). An additional method was employed to further validate these results. Taking advantage of the aforementioned differences in temporal compression between cue and trajectory ensembles, we conducted non-negative matrix factorisation to embed the temporal compression profiles of ensembles into a reduced features space (Supplementary Fig. 11a, b). The resulting projections were

assigned cue and trajectory class labels by an unsupervised k-means clustering method, which showed a strong correspondence with the labels assigned by our selection criteria (Supplementary Fig. 11b, c). Nevertheless, the projected density varied smoothly and continuously over this features space (Supplementary Fig. 11b). Therefore, although ensembles can be approximately discriminated based on their tendencies for encoding cues or trajectories, they are likely to exist, in actuality, over a continuum defined by the conjunctions between these two behavioural features. However, in consideration for conciseness and interpretability, we will treat these ensembles as belonging to two distinct classes for the remainder of this article.

### Reactivation of cue precedes trajectory around SWRs

Having established two separate forms of reactivated features, we explored whether cue and trajectory ensembles interact differently with the hippocampus around SWRs (Fig. 4a; Supplementary Fig. 12d). We found that, from the onset of SWR events, cue ensemble reactivations preceded trajectory ensemble reactivations during REST2 (Fig. 4b). Similarly, from the onset of reactivation events, ripple-band power peaked at a later time during cue reactivations compared to the reactivation of trajectories (Fig. 4d). Cross-correlation analysis between the reactivation strength time vectors of cue and trajectory ensembles revealed that cue ensemble reactivations preceded trajectory ensemble reactivations by an average of 126 ms (Fig. 4c), while ripple-band power during cue ensemble reactivation also lagged 238 ms behind the spectral power during trajectory reactivations (Fig. 4e). It is important to note here that, given our image sampling rate of ~19 Hz, the Galvo-Resonant scanners spent an average of ~50 ms of dwell time on the tissue for each frame. Therefore, the values of time delay reported here should be interpreted with a confidence interval of $\frac{50}{3} \approx \pm 17$ ms, corresponding to the theoretical average delay between when two neurons were scanned along the slow Galvo-axis. In contrast to REST2, a less clear relationship could be determined about REST1 activities. On the one hand, no significant temporal offset was observed between cue and trajectory reactivations from the onset of ripple events (Fig. 4b, c). On the other hand, though an average delay of 263 ms separated the ripple-band powers during cue and trajectory reactivations, this effect was detected at a much lower level of statistical significance (Fig. 4d, e). Therefore, evidence for shifted temporal interactions between cue and trajectory reactivations were inconclusive with regard to REST1.

One potential confounding factor is that the reactivation of trajectories takes a longer duration, given that, in the hippocampus, the replay of place cells occurs as a temporal sequence[17,19]; hypothetically, the reactivation of cues could take less time as the constituent neurons share similar spatial tuning curves and therefore need not be organized in a temporal sequence (cf. Supplementary Fig. 9). As a result, the observed delay may be influenced by the inherent timing properties of the two ensemble classes. This was not the case, however, as the median durations of the reactivation events across all three ensemble classes were comparable at ~180 ms (Fig. 4g). To further validate this hypothesis, cross-correlation was conducted between the onset times of ensemble reactivations and those of SWRs (Fig. 4f), in which case the durations of reactivation would no longer pose a bias. This analysis confirmed the same pattern of delay between cue and trajectory ensembles during REST2. Overall, the majority of ensemble reactivations onsets occurred after the onset of SWRs, with a median delay of 213.2 ms (Supplementary Fig. 12a, b). In consideration for the continuum of cue/trajectory features, we modelled the timing differences between ensemble reactivation and SWR onsets, as a function of the likelihood ratios for ensembles' cue and trajectory encoding tendencies. It was revealed that stronger cue-bias were related to earlier reactivation onsets from SWRs, while this timing was progressively delayed with more trajectory bias (Supplementary Fig. 12c). Moreover, the fraction of cue and trajectory reactivation events that were

associated with a SWR were comparable at 18.52% and 18.22% medians respectively in REST2 (Fig. 4h). However, a greater fraction of trajectory reactivation events were coupled with SWRs as compared to cue reactivations during REST1 (16.10% for cue and 21.93% for trajectory). In summary, these results indicate that awake reactivations of cue information following exposure to a spatial task tended to precede reactivations of trajectory information.

### Shared features in concurrent cue/trajectory reactivation

The timing difference observed between cue and trajectory reactivations in relation to SWRs suggests that, within the same recording session, some pairs of cue-trajectory ensembles may show significant temporal relationships. To identify these interactions, we performed cross-correlations between the reactivation strength time vectors across all cue-trajectory ensemble pairs concurrently recorded within the same session (Fig. 5a, b). Out of these ensemble pairs, 15.09% (24 pairs) expressed significant temporal coupling. Consistent with the cue-trajectory delay around SWRs, the average time lag between these coupled pairs, weighed by the cross-correlation coefficients, was 93.5 ms (Fig. 5c). We reasoned that these temporally coupled ensemble pairs would share certain functional features that distinguish them from the uncoupled pairs (Supplementary Fig. 13). As a preliminary step in visualizing these potential relationships, we grouped cue ensembles into three categories based on the cue location they reactivated most strongly towards (the last two cues were grouped under the same label on account of their spatial proximity). The average reactivation strength over spatial locations in the associated trajectory ensembles appeared to follow more closely the location(s) of the cues in the temporally coupled pairs, compared to the uncoupled group (Fig. 5d; Supplementary Fig. 6). This tendency was tested by taking the Pearson correlation between the features reactivated by cue-trajectory ensemble pairs (Fig. 5e). A two-way ANOVA model indicated that temporally coupled ensemble pairs shared a significantly higher degree of similarity in their reactivated features than uncoupled pairs. For REST1 ensembles, 22.22% (8 pairs) exhibited significant temporal coupling (Supplementary Fig. 14a). However, no difference was found in the similarity of reactivated features between coupled and uncoupled pairs (Supplementary Fig. 14b). With the current sample size, this result was inconclusive as it may reflect a feature of REST1 as well as a deficiency in statistical power to discern an effect.

Drawing inspirations from the mechanisms of pattern completion, we further tested this relationship using a Hopfield network model (Supplementary Fig. 15a). Sampling still from within the same recording sessions, we trained the network to learn two reactivated trajectory features, one that was coupled with a cue ensemble and one that was not. We then presented the corresponding reactivated cue feature as a partial retrieval pattern and tested for which of the two learned representations was retrieved. This procedure was conducted over all possible combinations in sessions that satisfied the training requirements. In 73.08% of these cases, the network successfully retrieved the coupled trajectory information (Supplementary Fig. 15b). Analogously, the Hamming distance between coupled cue and trajectory ensemble features were shorter compared to features reactivated by uncoupled pairs (Supplementary Fig. 15c). Taken together, these results demonstrated that cue and trajectory ensembles that expressed interlocked timings were likely to reactivate for complementary features of previous experience.

### Cue features exhibit higher stability across days

Lastly, we investigated whether the representations supported by cue and trajectory ensembles neurons express different degrees of stability over time. In 10 experimental animals, the same field-of-view had been imaged across consecutive recording days (Supplementary Fig. 16; Supplementary Table 1). We isolated spatially-selective cells that were part of cue or trajectory ensembles in REST2 within each

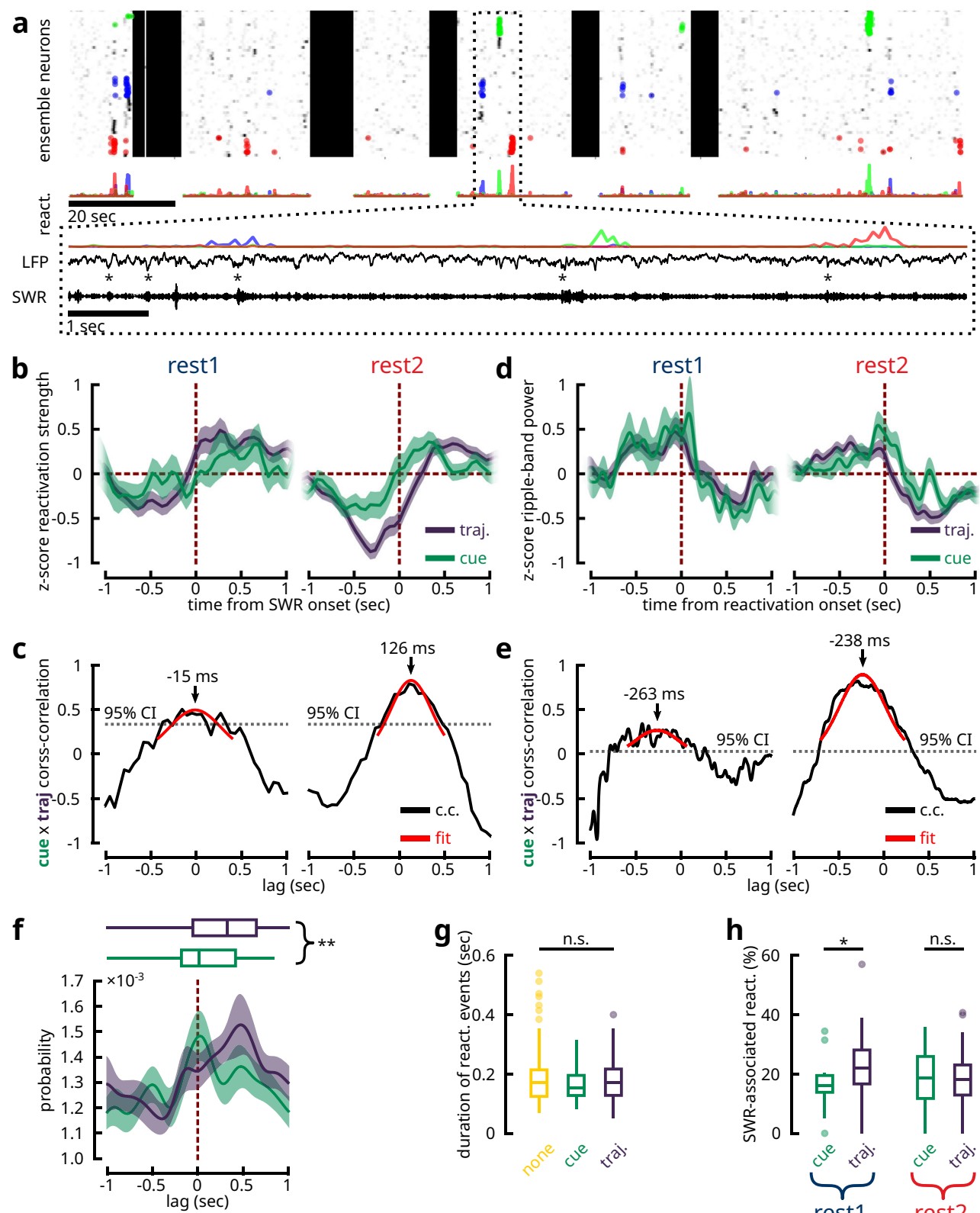

imaging session, and looked for overlapping ROIs on the subsequent recording day. We observed that cue ensemble neurons maintained a stable spatial tuning profile on the following test day, while trajectory ensemble neurons saw a higher degree of remapping (Fig. 6a–d). A large density of stable trajectory ensemble neurons coincided with the locations of landmarks, suggesting that they may be cue-responsive cells organized into trajectory sequences (Fig. 6a, b).

Given the tendency for cue features to be more stable, we reasoned that the same cue ensembles may be recruited during resting state across days. To test this possibility, we quantified the proportion of overlapping neurons between ensembles across days for all possible combinations of REST1 and REST2 (i.e., between REST1 ensembles across days, between REST2 ensembles across days and between REST1 ensembles and REST2 ensembles across days) (Supplementary

**Fig. 4 | Cue reactivations precede trajectory reactivations in relationship to hippocampal SWRs. a** A 3 min segment of a recording with simultaneous two-photon and hippocampal LFP acquisitions. Three distinct ensembles are colour-coded with their reactivation events highlighted by dots (same as Fig. 2a). The reactivation strength over time for each ensemble is shown. Zooming in on an 8 s window, consecutive reactivations of the three ensembles can be observed, whose occurrence coincided with SWRs. The reactivation strength, the broadband LFP (downsampled to 2.6 kHz) and the bandpass filtered SWR traces (150-250 Hz) are shown. The onset times of SWRs are labelled by asterisks. **b** Average z-scored reactivation strength of trajectory and cue ensembles centred at the onset of SWRs. Shaded areas denote the mean ± s.e.m. **c** Unbiased cross-correlation coefficients between cue and trajectory average reactivation strengths (**b**) as a function of lag period. Dotted line denotes the one-tailed 95% confidence interval. The coefficients were fitted to a Gaussian function to estimate the peak delay period. **d, e** Same as **b, c**, but for the average z-scored ripple-band power centred at the onset of ensembles reactivations. **f** Cross-correlograms between the onset time-stamps of cue and trajectory ensemble reactivations and the onset time-stamps of SWR events during REST2. Shaded area delineates 95% bootstrapped confidence intervals. Boxplots summarize the peak lag times in the cross-correlations (n = 59 cue, n = 118 trajectory ensembles; two-tailed Mann-Whitney U-test p = 0.00268). **g** The mean duration times for the reactivation of different ensemble classes show no significant difference across categories (n = 62 cue, n = 164 trajectory, n = 252 none REST2 ensembles; Kruskal-Wallis One-way ANOVA p = 0.6421). **h** Percentage of reactivation events associated with a SWR (n = 16 cue ensembles and n = 78 trajectory ensembles in REST1; n = 59 cue ensembles and n = 118 trajectory ensembles in REST2). During REST1, a greater number of trajectory reactivations were associated with SWRs (two-sample two-tailed Kolmogorov-Smirnov test p = 0.010; two-tailed Mann-Whitney U-test p = 0.036), while the same proportions were reported in REST2 (two-sample two-tailed Kolmogorov-Smirnov test p = 0.189; two-tailed Mann-Whitney U-test p = 0.638). All box plots show the median (line), the first and last quartiles (box), the minimum and maximum values (whiskers) and the outliers (circles). Source data are provided as a Source Data file.

Fig. 17a–d). During REST2, there was a slight tendency for similar cue ensembles to be found on the subsequent recording day, more often so than trajectory ensembles (Supplementary Fig. 17d, e). However, no difference in persistence was found across other REST combinations (Supplementary Fig. 17a–c, e). Overall, cue and trajectory ensembles expressed a similar degree of persistence across days, with an average of 22.73% and 20.59% persistent ensembles respectively (Supplementary Fig. 17f). Despite this lack of preference, REST2 contained a substantially higher fraction of persistent cue and trajectory ensembles across days (Supplementary Fig. 17g, h).

Within the same session, we found no preference between the recruitment of cue and trajectory ensembles across resting periods either (Fig. 6e), with 22.85% and 26.22% of cue and trajectory ensembles being reported as persistent respectively (Fig. 6f). Overall, these results indicate that cue ensembles consist of neurons that encode for more stable features across time, while trajectory ensemble neurons are more susceptible to remapping. However, this persistence in encoded features did not translate into a persistent recruitment of cue ensembles across recording days or across rest periods. Therefore, the process that determines the membership of offline ensembles is likely unbiased by functional features. Following locomotion, similar resting state ensembles are more likely to be found across days. Taken together, these results indicate that a recent exposure to a familiar experience invokes the reactivation of similar groups of neurons, despite the remapping of certain features.

## Patchy topographic organization in resting state ensembles

Lastly, we were interested in whether cue and trajectory ensemble neurons express differences in topographic arrangements. Naïvely, because sensory information tend to be organised in a topographic fashion in primary cortices (including in the motor cortex)[36], while the outflow of spatial information from the hippocampus would likely be distributed without discernible topography (cf. ref. [37]), we hypothesized that cue ensembles may exhibit a higher level of topographic clustering, whereas trajectory ensemble neurons may be more diffuse. We began by examining the distribution of cells' tendencies for encoding either cue or spatial locations, for all spatially-selective neurons. In this map, we observed a few patches of heightened densities of cells encoding for either feature (Supplementary Fig. 18a, b). However, no clear organization could be distinguished, suggesting that the topographic arrangement may be localized to patches. Four 100 × 100 μm windows were drawn over these patches (Supplementary Fig. 18b). Neurons found within the two windows over the regions with a higher prevalence of position-correlated responses form a uniform representation of spatial locations, with population vectors decorrelating smoothly over distance. In contrast, neurons contained within the two regions exhibiting preference for cue-responses form a

discontinuous representation biased by the locations of cues. Overall, these results suggest that a patchy topographic arrangement may bias the tuning tendencies of secondary motor neurons towards cues and positions.

Next, we examined the topographic distributions of ensemble neurons for different categories of ensembles (Supplementary Fig. 18c). Overall, cue ensemble neurons were more likely to be found over the lateral aspect (similar to Supplementary Fig. 18a), compared to trajectory ensemble neurons and neurons of unclassified ensembles, which were uniformly distributed (Supplementary Fig. 18d). To quantify the difference in these topographic distributions, the Kullback-Leibler divergence was taken between the probability density maps of each ensemble class and the density map of all neuronal ROIs (Supplementary Fig. 18e). Under this measure, the distribution of cue ensemble neurons was orders more different from the overall distribution of ROIs, compared to unclassified and trajectory ensemble neurons. An immediate hypothesis that can be generated from these results is that neurons that are part of a cue ensemble should be more clustered topographically due to their lateral confinement, whereas trajectory ensemble neurons should be more dispersed, assuming a random recruitment strategy for ensemble neurons. This was, however, not the case, as both cue and trajectory ensembles exhibit a similar degree of clustering in topographic space (Supplementary Fig. 18f, g). Nevertheless, over ~70% of ensembles of either category showed significant clustering, as opposed to dispersed. Therefore, resting state ensembles in M2 appear to be intrinsically organised into topographic patches. Taken together, these results suggest that both the distribution of cue-tending and position-tending cells, and the recruitment of resting state ensemble neurons follow a patchy arrangement in topographic space, though neither cue nor trajectory representations have a well-defined topographic organization.

## Discussion

We found that, following a virtual spatial navigation task, two distinct types of experience-related representations were reactivated in the motor cortex. On the one hand, synchronous ensembles of neurons during resting state consisted of cells that responded to visuo-tactile landmarks. On the other hand, short segments of continuous trajectories in space were formed from other sets of synchronous neuronal ensembles. Around the onset of SWRs detected in the ipsilateral hippocampus, the reactivation of cue information, on average, preceded that of trajectory information. The same pattern of delay was found between cue-trajectory ensemble pairs within the same recording sessions, whose patterns of reactivation showed significant temporal coupling. Such ensemble pairs reactivated for related features of previous experiences, whereby reactivated trajectories tended to occur around the locations of reactivated cues. Our findings illustrate a

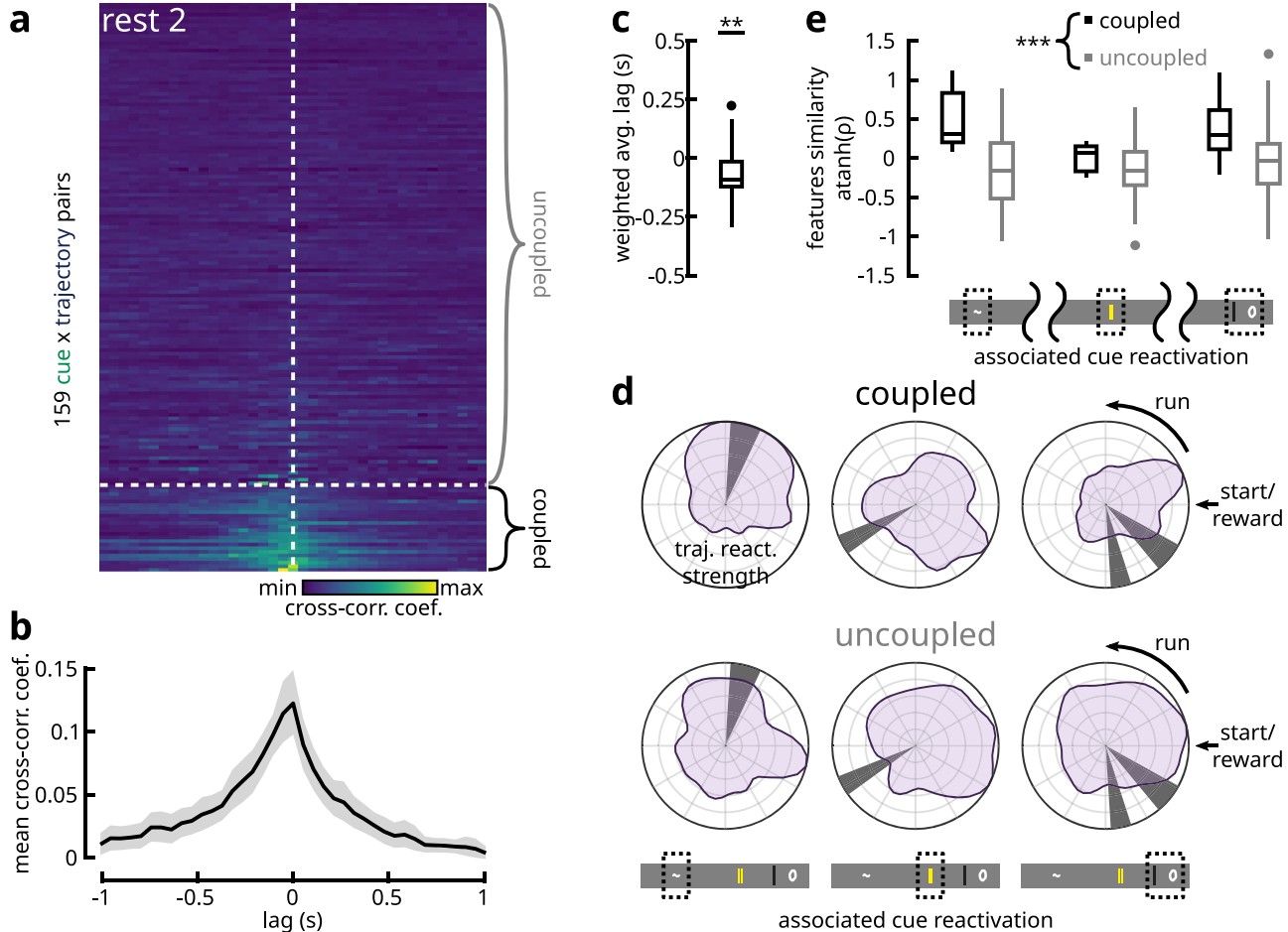

**Fig. 5 | Temporally coordinated cue-trajectory ensemble pairs reactivate for similar features of previous experience. a** Cross-correlation coefficients over a ± 1 s window between the reactivation-strength time-course vectors of cue-trajectory ensemble pairs found within the same REST2 session. Out of 159 such pairs, 24 showed significant temporal coordination. These pairs were labelled as coupled, whereas the remaining non-interacting pairs were referred to as uncoupled. **b** Average correlation-coefficient over time lags for coupled cue-trajectory ensemble pairs. Shaded area denotes bootstrapped 95% confidence intervals. **c** Average time lag, weighted by the cross-correlation coefficient, between coupled pairs suggested a significant delay from the reactivation of cue ensembles to that of the trajectory ensembles ($n = 24$ coupled pairs; two-tailed Wilcoxon sign rank test for the null hypothesis of zero median; $p = 0.0079$). **d** We separate the ensemble pairs into three groups, based on the cue location that the cue ensemble was most strongly reactivating for (the last two cues were labelled under the same group due to their spatial proximity). The average reactivation strength of the associated

trajectory ensembles as a function spatial location are illustrated in polar coordinate space. Qualitatively, the reactivated trajectory features tend to follow the associated cues more faithfully in the coupled pairs compared to the uncoupled pairs. **e** Pearson correlation coefficients between the reactivated cue and trajectory features in coupled and uncoupled pairs (atanh-tranformed for normality; for coupled pairs, at each cue from left to right, $n = 7, 10, 7$; for uncoupled pairs, $n = 17, 44, 74$). Coupled pairs expressed a greater degree of similarity in reactivated features than uncoupled pairs (two-way type II ANOVA with cue and temporal coupling as between factors; no significant interactions between factors; $p = 0.2299$; no effect of cue group on feature similarities; $p = 0.1291$; significant effect of temporal coupling on features similarity; $p = 5.562 \times 10^{-4}$; Shapiro's normality test $p = 0.2754$; Levene's equality of variances test; $p = 0.1736$). All box plots show the median (line), the first and last quartiles (box), the minimum and maximum values (whiskers) and the outliers (circles). Source data are provided as a Source Data file.

functional and timing-dependent relationship surrounding the offline retrieval of memories in cortical structures.

The existence of two parallel functional representations in the secondary motor cortex had been alluded to previously by multiple studies. In terms of visuo-tactile responses, the M2 is reciprocally connected with the somatosensory cortex as well as the visual cortical areas[38,39]. Accordingly, M2 has been found to respond to tactile stimulations in rodents[40], while lesioning the structure evokes somatosensory neglect[41]. Interestingly, optogenetic inactivation of M2 fibres projecting to S1 during NREM sleep caused impairments in a novel object recognition task requiring discrimination of tactile textures[42]. This finding demonstrated M2's involvement in the consolidation of somatosensory features. With regard to spatial coding, a number of recent articles have identified neuronal responses similar to those of place cells in multiple regions of the neocortex[5,20,31,43–46]. Crucially, the

formation of these representations was severely disrupted (including in M2) following bilateral lesions to the dorsal hippocampus[31,47]. These findings suggest that the hippocampal outflow of spatial information may pervade widely distributed regions of the neocortex.

Importantly, we found that both forms of representations were reactivated in M2 during quiet-wakefulness. A majority of previous investigations into cortical patterns of reactivation have uncovered features describing singular dimensions of behaviour, usually those related to the known functions of the structure. To name a few examples, it had been reported that the prefrontal cortex reactivated for task rules[12], the posterior parietal cortex for egocentric parameters[8], the retrosplenial cortex for environmental landmarks[11] and the primary motor cortex for movement sequences[10]. From these findings, one might conclude that cortical reactivations recapitulate aspects of previous experiences specific to the modality of the region.

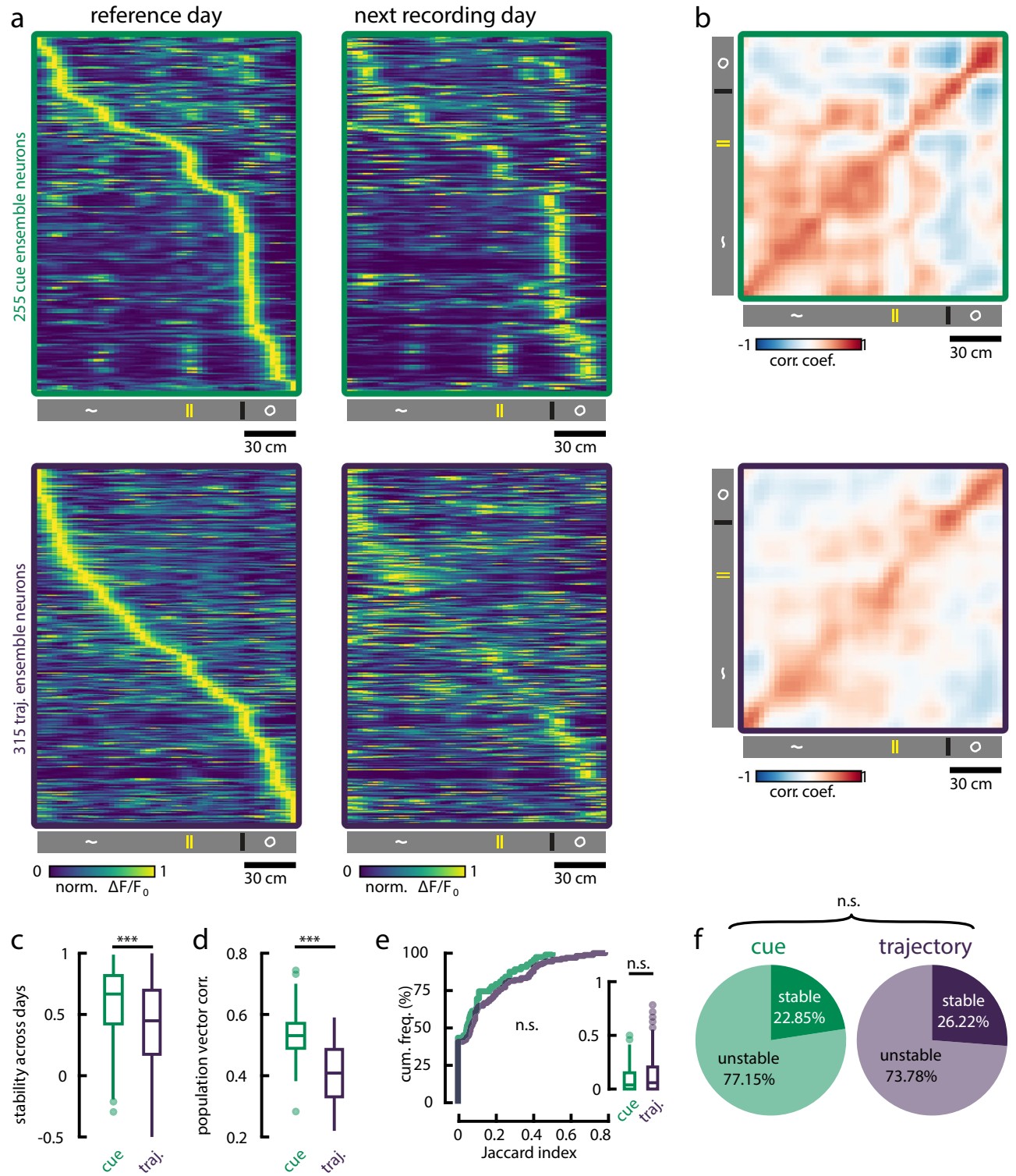

However, there are a few studies that demonstrated reactivation of spatial features in extra-hippocampal regions, namely in the visual[5], the posterior parietal[20], the medial prefrontal[6,18] cortices, the deep layers of the medial entorhinal cortex[7] and the ventral striatum[48], where the reactivation of such features were interlocked with hippocampal reactivations of the same behavioural experiences. Our current results help to reconcile these separate reports by suggesting that spatial and non-spatial features may be jointly expressed in the same cortical structure during offline periods. Nevertheless, given the limitations of the head-fixed preparation, it is hard to determine the exact

behavioural states the animals were in during the resting periods. Further research is needed to describe whether these patterns of reactivation express different dynamics during sleep or during particular states in awake quiescence.

Lending support to the presence of such a dichotomy is the temporal delay observed between the reactivation of cues and trajectory information. Classifying the reactivating ensembles into two categories based on their encoded features revealed that cue information was reactivated earlier on average than trajectory information. This finding complements a previous report in which patterns of

**Fig. 6 | Cue ensemble neurons support more stable representations across days. a** The same FOV was recorded over consecutive experimental days in 10 animals. Spatially-selective neurons that were part of cue and trajectory ensembles during REST2 were identified in the following recording day. Neurons' average activities as a function of position were sorted by the location of peak activity on the reference day. The same sorting was kept for tunings on the subsequent recording day. **b** Pearson correlation matrices of population vectors over positions (columns in **a**) between the reference day (y-axis) and the following recording day (x-axis). **c** Pearson correlation coefficients of neurons' spatial tuning profiles (rows in **a**) across recording days. Cue ensemble neurons maintained more stable tunings over days (one-tailed Mann-Whitney U-test; $p = 4.951 \times 10^{-9}$; $n = 255$ cue ensemble neurons; $n = 315$ trajectory ensemble neurons). **d** Pearson correlation coefficients of population vectors (columns in **a**) across recording days. Populations vectors express a higher degree of similarity across days for cue ensemble neurons (paired-sample one-tailed Wilcoxon signed rank test; $p = 4.131 \times 10^{-9}$; $n = 50$ spatial bins). **e** The Jaccard index was computed to measure the persistence

of ensembles between REST1 and REST2. For each REST2 ensembles, we calculated the number of member neurons overlapping with each REST1 ensembles. The Jaccard index was given as $J = \frac{O}{A + B - O}$, where for $O$ overlapping members, there are $A$ and $B$ total members in the REST1 and REST2 ensembles respectively. For each REST2 ensemble, we keep the maximum Jaccard index value following paired comparisons with all REST1 ensembles. Cue ($n = 62$) and trajectory ($n = 164$) ensembles were equally persistent across resting states (two-sample two-tailed Kolmogorov-Smirnov test $p = 0.725$; two-tailed Mann-Whitney U-test $p = 0.340$). **f** Following on **e**, we tested for the significance of the proportion of overlap by one-tailed Fisher's exact test against the null hypothesis that the overlapping fraction is not higher than chance. P-values lower than $\alpha = 0.001$ were considered as persistent ensembles. In REST2, a similar fraction of cue and trajectory ensembles remained persistent from REST1 ensembles (two-tailed $\chi^2$ test; $p = 0.574$; $\chi^2 = 0.3159$; df = 1). All box plots show the median (line), the first and last quartiles (box), the minimum and maximum values (whiskers) and the outliers (circles). Source data are provided as a Source Data file.

activity in the auditory cortex during NREM sleep preceding SWRs could accurately predict the ensuing CA1 activities, while these CA1 patterns could in turn explain the subsequent activities in the auditory cortex[9]. In that study, the offline activities in the auditory cortex were explained by distinct patterns of sounds, though the exact nature and contributions of hippocampal activities with regard to the cortex could not be determined. Here, our results suggest that the early and late reactivations surrounding SWR events may in fact constitute distinct functional representations. Given this temporal dissociation, it may be hypothesized that the earlier reactivation of cue-centric information reflect locally-encoded attributes in the secondary motor cortex, whereas the later reactivation of trajectory sequences may be driven by hippocampal outflow (cf.[29]). This notion is corroborated by the heightened stability of cue representations across recording days compared to trajectory information, which is consistent with the higher rates of synaptic turnover in hippocampus compared to neocortex[49].

The idea that local cortical patterns could serve as triggers for the retrieval of mnemonic information from the hippocampus places under scrutiny the potential for other cortical regions in contributing to this dialogue. In fact, rodent studies have demonstrated that specific auditory stimuli presented during NREM sleep could bias the content of hippocampal replay[9,50], while the presentation of olfactory cues had the same effect in a human fMRI study[51]. These sensory modalities, with the addition of visuo-tactile representations elaborated presently, imply that widely distributed neocortical sites could each serve as the initiator for hippocampal reactivation, where the subsequently evoked hippocampal patterns would permeate global cortical modules linking together the different attributes associated with episodic experiences. This model could explain the prevalence of trajectory reactivations over cue reactivations in our data. In particular, reactivated trajectory patterns observed in the secondary motor cortex could have been initiated by a separate region of the brain.

From the output side, it may be inferred from the present results that the information received by cortical sites downstream from the hippocampus during offline reactivation comprises sequential place cells activations analogous to the patterns of replay observed directly in the hippocampus[15–17,19]. Similarly, previous reports on coordinated reactivations between the hippocampus and the cortex for the same behavioural experiences have largely involved spatial features[5–7,20]. This observation calls into question the relevance of the place-code for cortical processing and consolidation. One possibility asserts that the hippocampal place-code encompasses a diverse range of non-spatial information as well[14,52,53], which is reflected in the modulation of firing rates within otherwise stable place fields by varying sensory/task conditions, a phenomenon known as rate remapping[54,55]. As such, the place-code provides a vehicle through which arbitrary associations could be formed between the diverse attributes of episodic

experiences spanning disparate modalities. In the present study, cue and trajectory ensembles that expressed temporally coordinated reactivations encoded for similar attributes of a recent experience. It is therefore possible, hypothetically, that the overlap between the spatial and the non-spatial dimensions is the semantic link that allows cortical cue features to be used to retrieve hippocampal patterns[30,56,57] (Fig. 7). Specifically, this retrieval process can be facilitated by pattern completion mechanisms supported by the CA3 recurrent network, which forms the basis of a content-addressable memory system[1,2,4,58,59]. The hippocampal output in turn elicits previously associated attributes of experience found across wide regions of the neocortex. This globally-coherent retrieval would hence allow cortico-cortical associations to form in such a way as to extract the statistical regularities found in those experiences.

This proposed hypothetical model has implications for both modulating online behaviour and mediating offline mnemonic processes. On the one hand, the targeted outflow of hippocampal spatial information onto cortical sites may provide a contextual and/or mnemonic frame of reference for guiding active behaviours[60]. Particularly, in the M2, it has been shown that population activities convey spatial/contextual information with a putative involvement in guiding action and decision-based planning[61]. Similarly, the interactions between the medial prefrontal cortex and the hippocampus appear important for navigation tasks that rely on memories of past trajectories[62]. On the other hand, the offline reactivation of recent experiences may drive changes in sensory encoding and functional representations in cortical areas. A recent study has shown that reactivation events in the lateral visual cortex more faithfully replicated the sensory responses of neurons in the future, rather than the preceding stimulus-evoked responses[63]. In parallel, neurons in the primary visual cortex gradually acquire sharper discrimination between distinct grating patterns with training[64]. Such descriptions of integrative processes that occur during both online and offline periods have also been hinted by our data. In particular, additional modelling has suggested that the features encoded by resting state ensembles exist over a continuum where varying degrees of cue and trajectory information is expressed. This conjunctive aspect may not only reflect a gradual consolidation of spatial information into existing cortical representations (cf. ref. [65]), but also a mechanism for imparting contextual information to sensory representations to guide online behaviour. Moreover, the heightened stability of cue representations over days, as compared to spatial information, may indicate a difference in the rate of consolidation or experience-driven drifts in functional representations in the cortex. Overall, a hypothesis can be proposed, stating that spatial information integrates with non-spatial information in the cortex to both modulate online behaviour through conjunctive coding and drive representational changes during offline periods. It remains to be determined how the functional encodings in cortical neurons

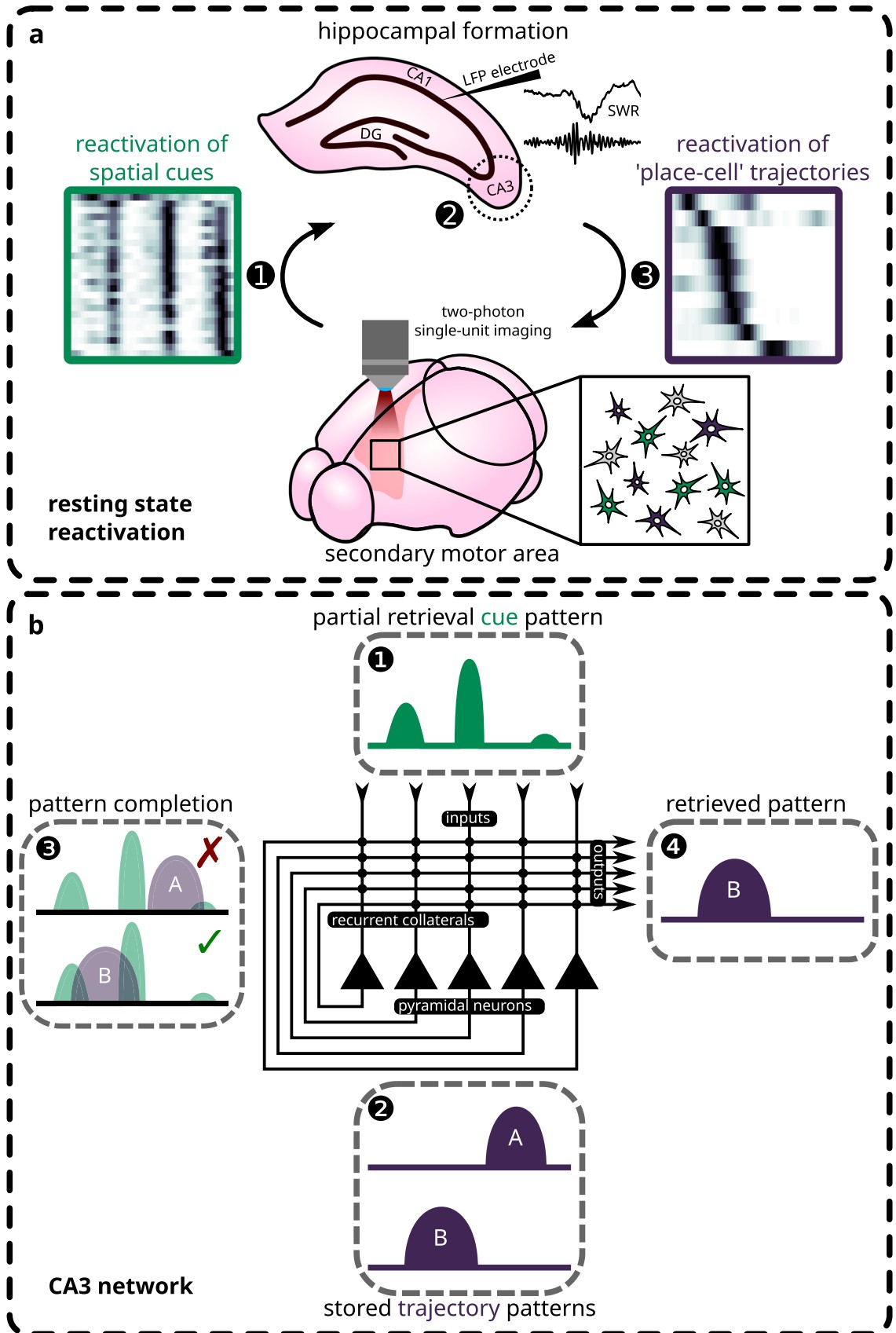

**Fig. 7 | Proposed hypothetical model for temporal dynamics of reactivation based on the theory of pattern completion. a** The temporal sequence that could account for the reported observations is illustrated. First, the secondary motor cortex spontaneously reinstates information related to visuo-tactile landmarks found within a recently explored virtual environment. Subsequently, these representations reach the hippocampal CA3 subfield, where pattern completion takes place in conjunction with sharp-wave ripples. Finally, the associated place cells trajectory sequence is retrieved and broadcast back to the cortex. **b** The putative computations that occur in CA3 during pattern completion are illustrated. Visuo-tactile landmark representations, inputted as partial retrieval cues to the CA3 recurrent network, are compared against the stored trajectory sequences from recent explorations inside a virtual environment. The trajectory pattern that shares the strongest similarity with the cued pattern is retrieved and outputted.

may be gradually shaped by the outflow of hippocampal place information and the degree to which this process may rely on the hippocampus.

Taken together, the current findings demonstrate a temporally organized sequence of cortico-hippocampal exchanges during offline reactivation, which is graded by the specific contents of the behavioural features being reactivated. These reactivated features likely arise separately from the hippocampus and the neocortex, and accordingly involves spatial and non-spatial aspects of previous experiences. As such, neocortical recollections of the specific 'attributes' of prior experiences may seed the reactivation of hippocampal sequences (i.e., episodes), which in turn is propagated back to neocortical regions. Pattern completion is proposed as a likely mechanism through which these separate dimensions of behaviour may be linked in order to facilitate the targeted retrieval of recent experiences from the hippocampus by the neocortex.

## Methods

### Animals and surgical procedure

All experiments were conducted in compliance with the guidelines established by the Canadian Council on Animal Care and were approved by the Animal Welfare Committee at the University of Lethbridge. Mice were single-housed under a 12 h light/dark cycle following surgery with mouse chow and water provided *ad libitum* until the beginning of experimentation. The room temperature is kept at 22 degrees Celsius and the humidity fluctuates between 40 and 60% depending on the season. A total of 14 Thy1-GCaMP6s transgenic mice were used (aged from 2–8 months old). All mice received a 5 mm cranial window implant over the dorsal cortex (AP: −3 to +2 mm from bregma; ML: centred on midline; Fig. 1b) following the same procedures as previously described[11,31]. 11 mice were also implanted with a bipolar electrode (0.5 mm tip separation; 50.8 $\mu$m Teflon-coated stainless-steel wires from A-M Systems) in dorsal CA1 stratum pyramidale ipsilateral to the imaging site. Electrodes were inserted from the posterior edge of the imaging window (AP: −3 mm; ML: 1.8 mm) with a 30° angle-of-approach along the AP axis pointed anteriorly. To determine the depth at which to lower the electrode, the signal was monitored through a speaker during lowering of the electrode. An abrupt increase in spikes signals that the lower tip has reached stratum pyramidale. Lowering the electrode by another ~0.5 mm should result in another burst of spikes, at which point the upper tip is within stratum pyramidale while the lower reference tip should be in stratum moleculare. Electrodes were implanted in the same hemisphere in which imaging was conducted. Two of the implanted electrodes showed poor signal quality, and the mice carrying these implants were excluded from electrophysiological analyses.

### Behavioural paradigm

Water-restricted mice were trained to run over a 150 cm long linear treadmill for a drop of sucrose water at the end of each lap, while under head-fixation (for training protocol and treadmill design, see[11,31,45,47]). The treadmill belt was lined with four distinct visuo-tactile cues (Fig. 1a) constructed out of hot glue (first and last cues), a strip of soft Velcro (second to last cue) and two strips of reflective tape (second cue). An LED light source was positioned in front of the animal to illuminate incoming cues. Before and after running, the treadmill was locked and the animals were habituated to rest quietly on the belt in addition to being trained to run. In most cases, consistent running and resting are learned after two weeks to a month of training. This typically required a week of clamping the belt for ~10 min before and after running. At first, animals will attempt to move the treadmill belt, but will soon learn to associate the absence of the reward port (moved away during rest periods) with immobility. With the belt clamped, the rotation encoder on the treadmill is still sensitive enough to detect motions from the animals, and these epochs were removed from subsequent analysis. Each imaging session proceeded with 10–20 min of resting (REST1), followed by ~8 min of running (RUN), and concluded with another 10–20 min of rest (REST2). Sessions in which animals ran less than 10 laps were discarded from further analysis.

### Two-photon imaging and ROI tracking over days

Imaging was performed under a Thorlabs Bergamo II microscope powered by a Ti:Sapphire femtosecond pulsed laser (Coherent Chamelion Ultra II) tuned to an excitatory wavelength of 920 nm and operated using the ThorImage 4.1 software. The light beam was rasterized by Galvo-Resonant scanners bidirectionally at a frame rate of ~19 Hz and focused onto the tissue via a 16 × water immersion objective (Nikon; NA = 0.8; 80–120 mW output power measured at the sample). Emitted light signals were amplified using a GaAsP photomultiplier tube (Hamamatsu) and digitized to a resolution of 800 × 800 pixels at 16 bit precision. The imaging FOV consisted of a 835 $\mu$m × 835 $\mu$m square plane acquired at a depth of 100-200 $\mu$m from the cortical surface to reach layers II/III. Depending on the imaging quality and the presence of bone-regrowth obstructing the window, the imaging window spanned anywhere between 0 mm and +1.67 mm over the AP axis and was centred on 0.5 mm ML (i.e., edging the superior saggital sinus; Fig. 1b).

Suite2p[66] was used to identify the neuron ROIs, and the extracted fluorescent traces were deconvolved by constrained non-negative matrix factorization[67]. The automatically detected ROIs were manually curated to remove any false-positives. In 10 animals, the same FOV was imaged across consecutive days, where specific landmarks such as blood vessels and neurons were used to guide the experimenter to manually align the FOV to that of the day-1 reference (Supplementary Fig. 16). To identify persistent neurons across two imaging sessions, the ROIs masks of the two session were binarized and registered against each other (Supplementary Fig. 19). Registration was constrained to rotation and translation only and was achieved through finding the nearest local minimum from the non-shifted images using a custom direct search gradient-free solver. ROIs that share 50% or more overlapping pixels were identified as the same neuron (percentage overlap calculated as the Jaccard index $\frac{O}{A+B-O} \times 100\%$, where out of $A$ and $B$ pixels contained by two candidate ROIs, $O$ pixels are overlapping).

Given the nature of the present study, opting for the secondary motor cortex over the somatosensory cortex would seem rather counterintuitive. The choice of imaging the secondary motor cortex is a practical one. Because the cranial window implant is designed to be attached flush against the dorsal aspect of the skull (our microscope does not rotate), we were limited to dorsal cortical regions. Out of these regions, only the hindlimb regions of the somatosensory cortex were accessible for imaging. This leads to a marked spatial delay in the

neuronal responses tied to visuo-tactile cues, which could bias subsequent analyses. By contrast, the cue-related responses in the secondary motor cortex were much more centred around cue locations. Furthermore, bone-regrowth tends to occur over the lateral edges of the imaging window. This affliction scarcely impacted the midline regions, away from the anterior edge of the window, in our experience. These experimental limitations led to the pragmatic consideration of imaging the secondary motor cortex.

### Electrophysiology and detection of SWRs
Local field potential was amplified 1000 × and band-pass filtered from 0.1–10,000 Hz through a Grass 7P122G amplifier. The analogue output was digitized using a Digidata 1550B unit sampling at 192 $\mu$s intervals (Axon pCLAMP 10 acquisition software). The signal was down-sampled to 2.6 kHz for analysis. To detect SWRs, local field potential was band-pass filtered from 150–250 Hz using a 400-order FIR filter. A RMS power envelope was extracted using an 8 ms sliding window. Amplitude regions exceeding 3 standard deviations from the mean were labelled as ripple events. 75% of this threshold was subsequently used to identify the times of onset and offset for each event. Ripple events must be at least 3 cycles long. Events that occurred less than 250 ms from a previous event were discarded (merged with the previous SWR).

### Detection of spatially-selective cells
The methods used for classifying neurons that express spatial encoding have been described extensively in previous works[11,31]. Briefly, two criteria must be met for successful identification. For the first criterion, the spatial information (SI) conveyed by a neuron about the animal's location must exceed the 95[th] percentile of a shuffled distribution. Spatial information was computed as[68]

$$I = \sum_{i=1}^{N} p_i \frac{f_i}{f} \log_2 \frac{f_i}{f}, \qquad (1)$$

where the average neuronal activity $f_i$ in the $i^{th}$ bin over the total average activity $f$, weighed by the spatial occupancy $p_i$, were evaluated over $N = 50$ spatial bins. The null distribution of SI was obtained by circularly shifting the time-course vectors of neuronal activities by a random factor 1,000 times. For the second criterion, neurons' place fields were identified by conducting a continuous wavelet transform over the spatial tuning curve of the neurons using a Ricker (Mexican Hat) wavelet. The scales evaluated were $\Sigma = \{1, 2, 3,...,50\}$ corresponding to the $N = 50$ spatial bins. Local maxima exceeding 3 median absolute deviations from the wavelet coefficients at the lowest scale of the transform ($\sigma = 1$) were identified as potential place fields. If a local maximum falls within the bounds of another maximum at a higher scale (i.e., wider place field), the candidate sitting at the lower scale (with a narrower place field) is discarded. The width of a place field must be between 5 and 80% of the total length of the environment. The mean activity within a place field must be 2.5 times higher than the activity outside of place fields. Peak activity during individual trials must occur within the place field in at least a third of the trials. Cells that supported at least one place field satisfying these constrained were determined to be spatially-selective.

### Detection of resting-state ensembles
Agglomerative clustering was used to detect groups of neurons that expressed highly synchronized activity during quiet wakeful periods (Fig. 2a) as previously described[11]. First, the time-courses of simultaneously recorded neurons were Gaussian smoothed ($\sigma = 200$ ms) in order to combat temporal jittering (cf. Supplementary Fig. 20) and to increase correlation between neuron pairs. Then, the Pearson correlation matrix between the z-scored time-courses of neuron pairs was taken. The correlation coefficients $r$ were converted into a distance

metric $d$ by $d = 1-r$. This value, which ranges from 0 to 2, describes how similar the time-course vectors between a pair of neurons are, with 0 being completely correlated, 1 being unrelated, and 2 being completely anti-correlated. Agglomerative clustering was then performed over this distance matrix using unweighted average distance criterion. A cut-off threshold of 0.75 was applied corresponding to an average correlation coefficient of 0.25 within each cluster of neurons. Clusters needed to contain at least 5 members in order to be classified as an ensemble.

### Classification of cue and trajectory ensembles
Irrespective of their final label, classification of ensembles began with screening for the presence of trajectories. Ensembles which contain less than 3 spatially-selective neurons were not considered. The normalized (from 0 to 1) spatial tuning curves (activity as a function of position) were extracted for each spatially-selective ensemble neuron. We identified continuous segments in space where the activity in the tuning curve of any ensemble neuron exceeded 0.5. Segments that were formed by less than 3 spatially-selective cells were discarded. These segments were also circularly wrapped for trajectories that crossed the starting/reward location. Out of these candidate trajectories, those with a length of less than 30 cm that spanned the centre of a cue were identified. Ensemble that carried such a segment were classified as cue ensembles, while the remaining ensembles that still contained a spatial segment were labelled as trajectory ensembles.

### Reactivation strength and extracting reactivated features
Hierarchical clustering permits the identification of neurons that are part of co-active ensembles, but otherwise does not provide information on the temporal activity of these ensembles. To further fine-tune the detected ensembles and to establish the time-courses of their resting-state dynamics, a custom PCA-ICA approach was devised (Supplementary Fig. 21). First, the deconvolved firing-rates matrix $\mathbf{X}$, with columns as individual neuron's time-course vectors, is normalized to have null mean and unitary variance. Typically, PCA and ICA extract a defined number of components, which are used as the basis vectors for a reduced space into which the firing-rates vectors are projected[12,69,70]. Here, the ensembles detected using hierarchical clustering serve as an initial estimate of these basis vectors defined in the matrix $\widetilde{\mathbf{W}}$. For each ensemble $m$, we define a column vector $\hat{\mathbf{u}}_\mathbf{m}$ of length $N$ in which the ensemble neurons are labelled 1 while the remaining neurons are 0. The vector is subsequently normalized to a unit vector $\mathbf{u}_\mathbf{m} = \frac{\hat{\mathbf{u}}_\mathbf{m}}{||\hat{\mathbf{u}}_\mathbf{m}||}$ pointing to the mean direction of the ensemble neurons:

$$\widetilde{\mathbf{W}} = \begin{pmatrix} \mathbf{u}_1 & \mathbf{u}_2 & \mathbf{u}_3 & \cdots & \mathbf{u}_M \end{pmatrix} = \begin{pmatrix} \frac{\hat{\mathbf{u}}_1}{||\hat{\mathbf{u}}_1||} & \frac{\hat{\mathbf{u}}_2}{||\hat{\mathbf{u}}_2||} & \frac{\hat{\mathbf{u}}_3}{||\hat{\mathbf{u}}_3||} & \cdots & \frac{\hat{\mathbf{u}}_M}{||\hat{\mathbf{u}}_M||} \end{pmatrix}. \qquad (2)$$

Note that $\widetilde{\mathbf{W}}^{\top} \widetilde{\mathbf{W}} = \mathbf{I}$ since there are no overlapping members between ensembles detected from hierarchical clustering. In other words, $\widetilde{\mathbf{W}}$ forms an orthonormal basis, just like the eigenvectors obtained from PCA. From here, projecting $\mathbf{X}$ into $\widetilde{\mathbf{W}}$ would yield the activity vectors over time for each ensemble defined as the mean firing-rate across all member neurons over time. However, doing so would assume that each neuron member of an ensemble contributes equally to the activation of that ensemble, which is unlikely to reflect the actual dynamics of the network given that separate pairs of neurons within the same ensemble share different degrees of temporal correlation. We will rely on the ICA algorithm to fine-tune this initial estimate of the basis, which will do so by optimizing over the quality of reconstruction so that the basis accurately captures the relationships structure in the temporal dynamics of the neuronal population.

First, de-noising is performed on $\mathbf{X}$ to limit the amount of drift in the basis vectors during ICA optimization caused by spurious relationships. This is achieved by subtracting the portion of the variance

captured by the initial basis estimate from $\mathbf{X}$:

$$\hat{\mathbf{X}} = \mathbf{X} - \mathbf{X}\widetilde{\mathbf{W}}\widetilde{\mathbf{W}}^{\top}. \qquad (3)$$

Then, PCA is conducted over $\hat{\mathbf{X}}$:

$$\text{corr}(\hat{\mathbf{X}}) = \boldsymbol{\Sigma}\boldsymbol{\Lambda}\boldsymbol{\Sigma}^{-1}, \qquad (4)$$

where $\boldsymbol{\Sigma}$ is the matrix of eigenvectors (principal components) and $\boldsymbol{\Lambda}$ is a diagonal matrix of eigenvalues such that $\text{diag}(\boldsymbol{\Lambda}) = \begin{pmatrix} \lambda_1 & \lambda_2 & \lambda_3 & \cdots & \lambda_N \end{pmatrix}$. The components whose associated eigenvalues are greater than the upper bound defined by the Marčenko-Pastur law $\lambda_+ = (1 + \sqrt{\frac{N}{T}})^2$ are kept[12,69]. Here, $T$ is the number of time bins in $\mathbf{X}$. In doing so, the variance in the data that is contributed by noise is removed and the remaining variance unexplained by the initial estimates is kept. Concatenating the initial basis estimate obtained from hierarchical clustering $\widetilde{\mathbf{W}}$ with the $L$ eigenvectors associated with the significant components obtained from PCA, $\boldsymbol{\Sigma}_{\lambda_+} = \begin{pmatrix} \boldsymbol{\sigma}_1 & \boldsymbol{\sigma}_2 & \boldsymbol{\sigma}_3 & \cdots & \boldsymbol{\sigma}_L \end{pmatrix}$, yields the final estimate of the transform matrix:

$$\widetilde{\mathbf{W}}' = \begin{pmatrix} \widetilde{\mathbf{W}} & | & \boldsymbol{\Sigma}_{\lambda_+} \end{pmatrix} = \begin{pmatrix} \mathbf{u}_1 & \cdots & \mathbf{u}_M & | & \boldsymbol{\sigma}_1 & \cdots & \boldsymbol{\sigma}_L \end{pmatrix}. \qquad (5)$$

Note that it can be shown that the matrix $\widetilde{\mathbf{W}}'$ is still an orthonormal basis; if $\lambda_1,\dots,\lambda_n$ and $\mathbf{v}_1,\dots,\mathbf{v}_n$ are the eigenvalues and eigenvectors respectively of the covariance matrix of the residuals in eq. (3), then $\mathbf{W}\mathbf{v}_i^{\top} = 0 \quad \forall \lambda_i > 0$.

The flavour of ICA used in the present study is reconstruction ICA[70]. This method was chosen over the popular FastICA algorithm for its higher computational efficiency (unconstrained optimization) and lower sensitivity towards un-whitened data. The original data matrix $\mathbf{X}$ is first projected onto the basis defined by the estimate transform matrix $\widetilde{\mathbf{W}}'$ to obtain $\mathbf{X}'$. Following this projection, only the variance accounted by the detected ensembles and the extra variance still remaining following PCA de-noising are kept. Then, reconstruction ICA is conducted by solving the following optimization problem:

$$\underset{\mathbf{W}}{\text{minimize}} \frac{1}{N}\sum_{i=1}^{N}||\mathbf{x}_i - \mathbf{x}_i\mathbf{W}\mathbf{W}^{\top}||_2^2 + \sum_{i=1}^{N}\sum_{j=1}^{k}g(\mathbf{x}_i\mathbf{W}_j), \qquad (6)$$

where $\mathbf{x}_i$ are the rows of $\mathbf{X}'$, $\mathbf{W}_j$ are the columns of $\mathbf{W}$ and $g(x) = \frac{1}{2}\log(\cosh(2x))$ is the contrast function that acts as a soft penalty term in place of the hard orthonormality constraint found in standard ICA. Here, the initial estimate of $\mathbf{W}$ passed to the solver is simply the identity matrix of size equal to the number of components in $\widetilde{\mathbf{W}}'$, given that the data has already been projected onto the basis of the estimate. Taking $\widetilde{\mathbf{W}}'\mathbf{W}^{\top}$ and extracting the first $M$ columns of the resulting matrix, corresponding to the number ensembles detected by hierarchical cluster, yields the final principal components. From here, the extraction of the reactivation strength time-course vectors, and the characterization of the reactivated features proceeds as originally described in[12]. That is, projecting $\mathbf{X}$ into this components space gives the reactivation strength as a function of time for each ensemble. Reactivation events were identified when the reactivation strength exceeds three standard deviations above the mean. The onset and offset of these events were delimited by 25% of this threshold. Similarly, to extract the features encoded by resting-state ensembles during RUN periods, the corresponding firing-rate matrix during RUN can be projected into this component space (Supplementary Fig. 22). Then, computing the mean ensemble activity over spatial locations gives the reactivated features.

## Reporting summary

Further information on research design is available in the Nature Portfolio Reporting Summary linked to this article.

## Data availability

The data generated in this study have been deposited in the DRYAD database under accession code https://doi.org/10.5061/dryad.1ns1rn91c[71]. Source data are provided with this paper.

## Code availability

The code that supports the findings of the present study is available on GitHub at https://github.com/LelouchLamperougeVI/OfflineEnsembles and deposited in the Zenodo database under the accession code https://doi.org/10.5281/zenodo.10030861[72].

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

## Acknowledgements

We thank Dun Mao, Rui C. Pais, Aubrey M. Demchuk, Hyun Choong Yong, Samsoon Inayat, Yagika Kaushik, Michael J. Eckert, Masami Tatsuno and David R. Euston for insightful discussions and comments. We thank Amanda Mauthe-Kaddoura, Karim Ali, Valérie Lapointe, Maurice Needham and Andrew McNaughton for technical assistance and logistics support. We thank JianJun Sun for surgical training and support. We thank Isabelle Gauthier, Di Shao and all members of the Animal Care Services and Animal Welfare Committee at the University of Lethbridge for making animal experiments possible. B.L.M discloses support for the research of this work from NIH Research Project Grant Program (R01) [no. NS121764], Defense Advanced Research Projects Agency (DARPA) [no. HR0011-18-2-0021], Natural Sciences and Engineering Research Council of Canada (NSERC) [no. 1631465] and Canadian Institutes of Health Research (CIHR) [no. PJT 156040]. M.H.M. discloses support for the research of this work from the Natural Sciences and Engineering Research Council of Canada, Canada Discovery Grant [no. 40352], Alberta Innovates, Alberta Alzheimer Research Program, Alzheimer Society of Canada, Alberta Prion Research Institute and Canadian Institute for Health Research. H.C. is supported by the Canada Graduate Scholarships for Doctoral Program (NSERC CGS-D).

## Author contributions

H.C., I.M.E., M.H.M. and B.L.M. designed the experiment. H.C., I.M.E. and A.R.N. conducted the experiments and collected the data. H.C. and I.M.E. performed the surgeries. H.C. analysed the data and wrote the manuscript, which all authors helped to revised.

## Competing interests

The authors declare no competing interests.
