## [Peer Review File · Nature Communications]

Cortical Reactivation of Spatial and Non-Spatial Features
Coordinates with Hippocampus to Form a Memory DialogueREVIEWER COMMENTS

Reviewer #1 (Remarks to the Author):

The mechanisms of episodic memory are not well understood. A key question is how the various features of experience (sight, sound, touch, spatial location) are linked together. This process requires interactions between the hippocampus and distributed cortical regions, and likely involves reactivation events whereby groups of neurons (ensembles) are reinstated during offline (post-experience) periods. This study by Chang et al. provides new insights into the nature of cortical reactivation by imaging populations of L2/3 cells in mouse secondary motor (M2) cortex during a REST-RUN-REST2 virtual navigation paradigm. They find that M2 neuron ensembles encoding either spatial or non-spatial (visuo-tactile) information are reactivated concurrently during the REST2 period. Moreover, they show that these different ensembles have temporally coordinated dynamics relative to each other and to hippocampal SWR events. They propose a model in which sensory-cue ensembles are first reactivated, this then triggers hippocampal-dependent reactivation of 'place cell like' ensembles.

Overall, this is a thought-provoking and nicely executed study that progresses our understanding of reactivation dynamics in cortex. In particular, the finding of coordinated reactivation of spatial/non-spatial ensembles in the same cortical region is a novel advance.

My major concern relates to the conceptualization of two discrete types of ensembles. Whereas it is nicely demonstrated that different M2 reactivation ensembles encode varying features (visuo-tactile, spatial), it's less clear that a hard binary distinction between them should be drawn. Indeed, in the Results (L256) the authors recognize these ensembles likely represent a continuum. They write: "For the remainder of this article, a clear distinction will be made between these two ensembles classes in consideration for conciseness and interpretability. However, it should be stressed that a spectrum likely exists over the degrees to which reactivating ensembles encode for either behavioural parameter". The argument is that it is helpful to dichotomize the ensembles for the sake of clarity, interpretation, narrative, etc. Still, I wonder about the appropriateness of this framing of two discrete ensembles, or whether it would be best to confront what appears to be a continuum of ensemble response properties from 'cue like' to 'place cell like' (Supplementary Figure 6a).

To deal with this question, I would recommend the authors consider several revisions to the manuscript.

- In the Discussion, it would be helpful to reiterate and expand on the idea there exists a continuum between the ensembles and to provide some additional interpretation regarding this.
- Perform some additional analysis that does not binarize the ensembles, but rather computes correlations between a metric describing an ensemble as 'cue like – place cell like' (e.g. likelihood ratio in SFig6a) and correlate this with other variables such as time of reactivation relative to SWR onset.

- Is it possible to perform unsupervised clustering on the ensembles? Does this reveal distinct groups?
- In an important previous paper from this group, they showed that hippocampal lesions particularly impacted spatial coding (Esteves et al. 2021). Did the authors measure reactivation events in these experiments? Are 'cue-like' ensembles specifically spared, whereas 'place cell-like' ensembles are disrupted?
- Are there any features of the two putative ensemble types that correlate with other independent variables, e.g. spatial distribution of ensemble in M2, cortical depth, spontaneous activity levels, etc. that might provide additional evidence that these are indeed distinct groups of neurons that are differentially embedded in the cortical circuitry?

Reviewer #2 (Remarks to the Author):

This manuscript by Chang et al describes their study on the reactivation of neuronal ensembles in the secondary motor cortex (M2) during resting in mice, after running on a treadmill with visual and tactile cues. Neuronal activities were recorded using calcium imaging during the running (RUN) and during the resting before (REST1) and after (REST2). The authors identified synchronous neuronal ensembles using a sophisticated computational method and quantified the spatial contents of neuronal members in the ensembles. The authors found two types of reactivated ensembles, cue and trajectory ensembles. The cue ensemble reactivation events preceded the trajectory ones and both were correlated with hippocampal sharp-wave ripples (SWRs). The authors concluded that a cortical-hippocampal-cortical communication loop enables the offline reactivation of distinctive cue and spatial memory components.

The manuscript presents fresh interesting data on an important question of how different mnemonic components are reactivated offline (and thus consolidated) in the cortical-hippocampal circuits in offline behavior. The area M2 is a good target, given that it receives sensory input and presumably generates an internal motor plan for spatial navigation. The experimental approach and analysis methodology are sufficiently described, although the computational method for ensemble identification is not easy to comprehend. The data are valuable for advancing the field of neural circuit mechanisms of offline memory consolidation. However, there are major concerns on the presented results and the main conclusion, as elaborated below.

1. Cue versus trajectory ensembles

A major finding is that the identified ensembles during REST2 contained two types of distinguishing contents: cue vs. spatial trajectory. However, this distinction is not clear to me at least from the data presented.

i) It seems that the two types were defined solely by a threshold of 30 cm in the span of aggregated spatial tuning activities among member neurons (Line #708). I am not sure whether this threshold warrants the distinction. Some presented raw examples (e.g. Fig.2a) seem to show cue ensembles were more tied to cue locations. However, their relationship with cues (distance to cue locations) seems not fixed (reliably) and not used for ensemble definition. More importantly, in these and, more obviously, in other examples (e.g. Fig. 1f), there were accumulations of cells at cue locations in trajectory ensembles and there were cells with spreading place fields in cue ensembles. It is unclear to me how clear cut these two types of ensembles were. Ideally, cue ensembles (cue cells) should be established by cue manipulations (e.g. moving cues along a trajectory). Alternatively, in the current dataset, are there measures of ensembles that clearly show a bi-modal distribution?

ii) The authors show that the two types ensembles differed in several properties, providing evidence for their distinction. However, some of these properties are either a direct result of their definition (Fig. 2d) or biased by the definition (Fig. 2c).

iii) In Sup Fig. 3, the authors presented data from their previous study to distinguish two type of M2 cells by their dependence on hippocampal lesion and to show that the hippocampal independent cells represent cues (rather than space per se). Whereas the lesion did result in a larger share of cells active around some cues (not all) in the raw plot, the quantified result (panel e) shows a similar increase in decoding error between on and off cue locations. It seems the lesion influenced cells at all locations. I am puzzled why the title of this figure says "Representation of visuo-tactile cues is preserved...".

2. Nature of neural ensemble activities in REST2

The manuscript presents convincing data that clusters of cells active at specific locations during RUN were also activated together during REST2. However, after reading the manuscript, it is hard to have a clear idea what the REST2 synchronous activities looked like and whether they were truly offline reactivation as claimed.

i) It is unclear what behavioral state the cortex was in REST1 and REST2. The presence of hippocampal SWRs is commonly regarded as evidence of an offline state, although it could occur when the animal stays awake (awake ripples or replay). Since local field potentials (LFPs) were not recorded from M2, it is unknown whether the cortex was in a state (with LFP marks) different from the active behavior in RUN. Although the authors removed the time periods when animals were actively moving in REST1 and REST2, the cortex could be still active even during immobility.

ii) The manuscript does not quantify the duration of the identified cue/trajectory ensembles. The examples in Fig. 3a seem to have durations about a second. This is quite long compared to previously

published cortical reactivation events. More importantly, how was the durations of ensembles in REST2 compared to those during RUN? Was there compression in the reactivation as previously shown?

3. Cortical-hippocampal-cortical loop

In the abstract and the discussion (Fig.6, Lines #577-583), the authors concluded that cue ensembles were reactivated in the cortex first. This cued the hippocampus to reactivate spatial representations, which in turn reactivated the cortical trajectory ensembles. I am puzzled how the conclusion was reached.

i) There are no data that support this temporal order of reactivation. If we assume hippocampal reactivation mostly occurred around SWRs, the presented data only show that both cue/trajectory reactivations in M2 occurred later than hippocampal ones (Fig. 3) and perhaps cue reactivation preceded trajectory reactivation (Fig. 4). But there are no data to show hippocampal SWRs were occurring between cue and trajectory reactivation events.

ii) The timing relationship plots in Figs. 3&4 show mostly broad peaks with long time delays (~hundreds milliseconds). It is understandable that the slow sampling rate (19 frames per second) limited the accuracy in determining temporal relations. Given that reactivation is commonly believed to be short events, it seems difficult to make the main conclusion from this type of data with limited temporal resolution.

Reviewer #3 (Remarks to the Author):

Over the past several decades, each wave of results on memory trace reactivation has further underscored its importance and how much remains unknown. In this report, Chang and co-authors take a major step toward understanding the dynamics of reactivation by examining relationships between two different kinds of neural ensembles within premotor cortex. The study is important, the dataset impressive, and the analyses are convincing and relatively thorough. These results will be important not only within the neural systems field but more widely to anyone interested in how the brain processes and consolidates information. With respect to weaknesses, the authors unexpectedly omit analysis of the spatial relationship between representation types on the cortical surface, which is relevant to the stated justification of examining different dimensions in the “same” cortical region. Additionally, several minor edits would improve clarity and interpretation.

Major points

1. The one major concern relates to the absence of information on how neurons within and between ensembles are distributed across the cortical surface. This information would be valuable for, first, understanding how anatomically distributed the representations are (are ensembles with more trajectory information more anatomically localized, or dispersed, than the "cue" ensembles?) and, secondly, for confirming that the respective pools of multi-selective cells are, in fact, in the same cortical region. Line 122 states "[T]he current study aimed to explore whether these distinct representations, which are likely of hippocampal and cortical origin respectively, are reactivated concurrently by the same cortical region." However, the size of the imaging window is large enough to sample multiple subdivisions of premotor cortex, raising questions about the extent to which the ensembles are in fact in the same "region".

Minor points

Introduction

- Line 113: "Given that diverse cortical areas encompass a wide range of cognitive processes, the functional links that would permit associations to be formed between the hippocampus and the cortex remains to be elucidated." Here "given" seems to be the wrong word (Why does the former imply the latter?)

Results

- Line 142 "the network, following locomotion, exhibited more stereotyped patterns of activity characterized by sparse population vectors". There are a greater number of ensembles, with on average more neurons each, why are these necessarily "more sparse" as population vectors? Why would this be hypothesized? Sparsity can mean many different things, and immediately after this statement the vectors are referred to as temporally sparse. It would be useful to unpack in each case what the authors mean by sparsity.

- Are differences between temporal sparsity during REST 1 and 2 meaningful, given the lack of apparent controls for sleep (or simply fatigue) and time over the rest intervals?

- Obviously there is no way to test whether REST 1 ensembles were also spatially selective for spaces that the mouse sampled before rest was recorded. It may be important to qualify that “spatial selectivity” refers specifically to selectivity of spaces in the behavioral apparatus. The claim that the ensemble neurons were “drawn at random from the sample population” seems unwarranted, particularly without an analysis of the anatomical distribution of the ensemble cells in REST 1 versus REST 2.

- Line 228 “Both trajectory and cue ensembles contained a significantly higher number of spatially-selective cells than the ensembles that were unclassified.” Wasn’t spatial selectivity required for ensembles to be classified as cue or trajectory? If that’s the case, then is this claim circular? (Ensembles were classified on account of spatial selectivity -> analysis showed that spatial selectivity was higher in classified ensembles)

- Line 237 “Trajectory ensembles were more than three times as prevalent”. How does this reflect the apparent difference in proportion of “cue” versus “trajectory” neurons? (Or, if difficult to classify as single neurons, the distribution of “cue-like” versus “trajectory like” using the model-fitting method.)

- In the section “Cue and trajectory ensembles reactivate for complementary features” it is not immediately obvious that pairs of “cue and trajectory” refers specifically to cue-trajectory pairings. On first read, this appeared to mean “pairs of cue ensembles, pairs of trajectory ensembles, and pairs of cue-trajectory ensembles”. Some change in the language (e.g., “cue-trajectory pairs”) would be useful to make this section easier to follow on an initial read.

Discussion

- The difference in day-to-day stability between cue and trajectory ensembles is very interesting, as it potentially offers a glimpse into how space/trajectory information is encoded and consolidated in M2 over learning. To put another way, the data hint at how contextual information may influence M2 activity specifically for some dimensions and not others. An alternative explanation for this result might be that the “partial remapping” of trajectory information over days relates to minor changes in movement behavior between cue locations, compared with “at” cue locations. Importantly, these two interpretations aren’t exclusive, and in fact could be related. Either way, it may be nice to see more elaboration on this result in the Discussion, ideally with some reference to work by Doug Nitz and colleagues illustrating how M2 may “transform spatial information into planned action during navigation” (Olson et al., 2020).

REVIEWER COMMENTS

Reviewer #1 (Remarks to the Author):

The mechanisms of episodic memory are not well understood. A key question is how the various features of experience (sight, sound, touch, spatial location) are linked together. This process requires interactions between the hippocampus and distributed cortical regions, and likely involves reactivation events whereby groups of neurons (ensembles) are reinstated during offline (post-experience) periods. This study by Chang et al. provides new insights into the nature of cortical reactivation by imaging populations of L2/3 cells in mouse secondary motor (M2) cortex during a REST-RUN-REST2 virtual navigation paradigm. They find that M2 neuron ensembles encoding either spatial or non-spatial (visuo-tactile) information are reactivated concurrently during the REST2 period. Moreover, they show that these different ensembles have temporally coordinated dynamics relative to each other and to hippocampal SWR events. They propose a model in which sensory-cue ensembles are first reactivated, this then triggers hippocampal-dependent reactivation of 'place cell like' ensembles.

Overall, this is a thought-provoking and nicely executed study that progresses our understanding of reactivation dynamics in cortex. In particular, the finding of coordinated reactivation of spatial/non-spatial ensembles in the same cortical region is a novel advance.

We thank the reviewer for highlighting the impact of our work and for their invaluable suggestions to help improve it. Please find below answers to your specific comments.

My major concern relates to the conceptualization of two discrete types of ensembles. Whereas it is nicely demonstrated that different M2 reactivation ensembles encode varying features (visuo-tactile, spatial), it's less clear that a hard binary distinction between them should be drawn. Indeed, in the Results (L256) the authors recognize these ensembles likely represent a continuum. They write: "For the remainder of this article, a clear distinction will be made between these two ensemble classes in consideration for conciseness and interpretability. However, it should be stressed that a spectrum likely exists over the degrees to which reactivating ensembles encode for either behavioural parameter". The argument is that it is helpful to dichotomize the ensembles for the sake of clarity, interpretation, narrative, etc. Still, I wonder about the appropriateness of this framing of two discrete ensembles, or whether it would be best to confront what appears to be a continuum of ensemble response properties from 'cue like' to 'place cell like' (Supplementary Figure 6a).

The reviewer was concerned about the appropriateness of drawing a sharp distinction between cue and trajectory ensemble classes when interpreting and conceptualizing the present work. As the reviewer pointed out, our evidence showed that the degree to which ensembles tend to encode cue vs trajectory features likely exists over a continuum of conjunctive features. As such, introducing some additional analyses that directly address this continuous property could help make the article more comprehensive and informative about the underlying functional processes. Towards this end, they suggested several analyses/revisions. Below are details on how we implemented these suggestions.

To deal with this question, I would recommend the authors consider several revisions to the manuscript.

- In the Discussion, it would be helpful to reiterate and expand on the idea there exists a continuum between the ensembles and to provide some additional interpretation regarding this.

We expanded the relevant sections in the Discussion to offer more complete perspectives (Lines 650-660).

- Perform some additional analysis that does not binarize the ensembles, but rather computes correlations between a metric describing an ensemble as ‘cue like – place cell like’ (e.g. likelihood ratio in SFig6a) and correlate this with other variables such as time of reactivation relative to SWR onset.

Using a robust linear regression, we modelled the relationship between the difference in onset times of ensemble reactivations and SWRs, and the ensembles’ tendencies for encoding cue vs trajectory information (provided as the likelihood ratio). We found that stronger bias for cue information was linked with earlier reactivation times from SWRs, while ensembles biased more strongly to trajectories reactivated later.

Changes:

- Results (Lines 343-347)
- Supplementary Fig. 12c

- Is it possible to perform unsupervised clustering on the ensembles? Does this reveal distinct groups?

We performed a temporal compression analysis in response to a different reviewer’s comment. Serendipitously, this analysis revealed a metric over which unsupervised clustering could be conducted to both evaluate our selection criteria and look for signs of bimodality in the features reactivated by ensembles. Briefly, we found that trajectory ensembles showed significantly more temporal compression compared to cue ensembles (Supplementary Fig. 9). This is consistent with the notion that cue ensemble neurons, as they are presumed to respond simultaneously to cue sensation, would also fire simultaneously during reactivation. In contrast, the trajectory sequences during RUN were estimated to be compressed ~30-fold during reactivation epochs. The measure of temporal compression can, therefore, be used as a metric for sequential activity. We conducted dimensionality reduction over the temporal compression profiles of ensembles using non-negative matrix factorization. Within this reduced features space, a continuous density was observed, consistent with the notion that ensembles’ tendencies towards encoding cue and trajectory information exist over a continuum. Nevertheless, unsupervised clustering of the ensembles by k-means yielded labels that corresponded well with the labels assigned using our criteria. Therefore, a rough distinction can be drawn between the two ensemble classes. Specific details about this analysis can be found in the following changes:

- Results (Lines 269-277)
- Supplementary Fig. 11

- In an important previous paper from this group, they showed that hippocampal lesions particularly impacted spatial coding (Esteves et al. 2021). Did the authors measure reactivation events in these experiments? Are ‘cue-like’ ensembles specifically spared, whereas ‘place cell-like’ ensembles are disrupted?

Regrettably, we did not record resting sessions in that study. As the reviewer pointed out, an immediate hypothesis from the current work, indeed, posits that reactivations of cue information would be spared following hippocampal lesion, whereas trajectory reactivations should be disrupted. Alternatively, trajectory reactivations could still persist in an environment familiar to

the animal prior to lesions; in a recent follow-up article (Esteves et al., 2023, iScience), we demonstrated evidence for the consolidation of “virtual” spatial memory traces in the cortex.

Actions taken:

- We offered this possibility as a future direction in the Discussion (Lines 662-663).
- See also newly added Supplementary Fig. 7

• Are there any features of the two putative ensemble types that correlate with other independent variables, e.g. spatial distribution of ensemble in M2, cortical depth, spontaneous activity levels, etc. that might provide additional evidence that these are indeed distinct groups of neurons that are differentially embedded in the cortical circuitry?

One of the main conclusions of this study was the reactivation of two different functional features, one spatial and one non-spatial. As such, a more thorough comparison between the two classes, over independent variables, would be necessary to adequately describe these two putative functional features and to tell whether they are indeed distinct.

Actions taken:

- Extra metrics have been added to further compare cue vs trajectory ensembles, including the mean delta-F/F, rate of calcium transients and rate of reactivations (Supplementary Fig. 8; Lines 235-241 in Results).
- Additional analyses that further characterize the differences between cue and trajectory ensembles have been included in response to other reviewer comments (Supplementary Fig. 9, 11, 18).

Reviewer #2 (Remarks to the Author):

This manuscript by Chang et al describes their study on the reactivation of neuronal ensembles in the secondary motor cortex (M2) during resting in mice, after running on a treadmill with visual and tactile cues. Neuronal activities were recorded using calcium imaging during the running (RUN) and during the resting before (REST1) and after (REST2). The authors identified synchronous neuronal ensembles using a sophisticated computational method and quantified the spatial contents of neuronal members in the ensembles. The authors found two types of reactivated ensembles, cue and trajectory ensembles. The cue ensemble reactivation events proceeded the trajectory ones and both were correlated with hippocampal sharp-wave ripples (SWRs). The authors concluded that a cortical-hippocampal-cortical communication loop enables the offline reactivation of distinctive cue and spatial memory components.

The manuscript presents fresh interesting data on an important question of how different mnemonic components are reactivated offline (and thus consolidated) in the cortical-hippocampal circuits in offline behavior. The area M2 is a good target, given that it receives sensory input and presumably generates an internal motor plan for spatial navigation. The experimental approach and analysis methodology are sufficiently described, although the computational method for ensemble identification is not easy to comprehend. The data are valuable for advancing the field of neural circuit mechanisms of offline memory consolidation. However, there are major concerns on the presented results and the main conclusion, as elaborated below.

We thank the reviewer for highlighting the impact of our work and for their invaluable suggestions to help improve it. Please find below answers to your specific comments.

1. Cue versus trajectory ensembles

A major finding is that the identified ensembles during REST2 contained two types of distinguishing contents: cue vs. spatial trajectory. However, this distinction is not clear to me at least from the data presented.

i) It seems that the two types were defined solely by a threshold of 30 cm in the span of aggregated spatial tuning activities among member neurons (Line #708). I am not sure whether this threshold warrants the distinction. Some presented raw examples (e.g. Fig.2a) seem to show cue ensembles were more tied to cue locations. However, their relationship with cues (distance to cue locations) seems not fixed (reliably) and not used for ensemble definition. More importantly, in these and, more obviously, in other examples (e.g. Fig. 1f), there were accumulations of cells at cue locations in trajectory ensembles and there were cells with spreading place fields in cue ensembles. It is unclear to me how clear cut these two types of ensembles were. Ideally, cue ensembles (cue cells) should be established by cue manipulations (e.g. moving cues along a trajectory). Alternatively, in the current dataset, are there measures of ensembles that clearly show a bi-modal distribution?

The reviewer is hesitant about whether resting ensembles indeed belong to two discreet categories, given that no sufficient evidence was provided to show a clear distinction in their encoded features. As the reviewer pointed out, cue-encoding cells should have, ideally, been identified experimentally by manipulating the cue locations or introducing sessions with a blank belt. This was, however, not the case in the present set of experiments. In the original version of the manuscript, we employed a model-fitting procedure to obtain a measure of ensembles' tendency to encode either cue or trajectory information (Supplementary Fig. 10). The distribution of this measure was continuous and showed no bi-modality. In response, we specified in writing that resting ensembles likely fall over a continuum of conjunctive features with varying degrees of cue/trajectory encoding tendencies: "Therefore, although ensembles can be approximately discriminated based on their tendencies for encoding cues or trajectories, they are likely to exist, in actuality, over a continuum defined by the conjunctions between these two behavioural features. However, in consideration for conciseness and interpretability, we will treat these ensembles as belonging to two distinct classes for the remainder of this article." (Lines 277-281). We acknowledge that this important property should have been stated more clearly and prominently to avoid confusion and misinterpretation. We made several revisions to address this issue:

- Extra analyses and metrics were introduced to further characterize the features encoded by resting ensembles and whether they are, indeed, separable on the basis of cue/trajectory information (Supplementary Fig. 8, 9, 11, 18) The collection of these results confirms our initial findings, i.e., ensembles exist over a continuum of conjunctive cue/trajectory information.
- The Discussion section was expanded to re-emphasize and elaborate on the continuous/conjunctive nature of resting ensembles in relation to cue and trajectory encoding (Lines 650-660).

ii) The authors show that the two types ensembles differed in several properties, providing evidence for their distinction. However, some of these properties are either a direct result of their definition (Fig. 2d) or biased by the definition (Fig. 2c).

The reviewer identified certain important biases in the metrics used to compare cue and trajectory ensembles. More independent measures are needed to tell whether these putative ensemble classes are indeed different.

Actions taken:

- Extra metrics have been added to further compare cue vs trajectory ensembles, including the mean $\Delta F/F$, rate of calcium transients and rate of reactivations (Supplementary Fig. 8; Lines 235-241 in Results).
- Additional analyses that further characterize the differences between cue and trajectory ensembles have been included in response to other reviewer comments (Supplementary Fig. 9, 11, 18).

iii) In Sup Fig. 3, the authors presented data from their previous study to distinguish two type of M2 cells by their dependence on hippocampal lesion and to show that the hippocampal independent cells represent cues (rather than space per se). Whereas the lesion did result in a larger share of cells active around some cues (not all) in the raw plot, the quantified result (panel e) shows a similar increase in decoding error between on and off cue locations. It seems the lesion influenced cells at all locations. I am puzzled why the title of this figure says “Representation of visuo-tactile cues is preserved...”.

The reviewer identified an inaccurate interpretation of results related to the population encoding of locations and cues following hippocampal lesions. Indeed, the Bayesian decoding results shown in panel e highlighted an increase in error for both on-cue and off-cue locations following lesions. This is corroborated by the lack of significant interaction between the main factors in the ANOVA analysis; one would expect a significant interaction if on-cue decoding was preserved following lesions. Instead, decoding error at cue locations appears to be reduced in both control and lesion groups, as seen in panel d as well. The collection of results shown in the figure suggest that the encoding of cues became more apparent at the population level, though no evidence of complete preservation is available.

Actions taken:

- The title of the figure was renamed to be more representative of the actual finding: “Representation of visuo-tactile cues in the secondary motor cortex is more pronounced following bilateral lesion of the dorsal hippocampus.”
- Corrections were made to the relevant sentences in the Introduction (Lines 77-78) and Results (Lines 201-203).

2. Nature of neural ensemble activities in REST2

The manuscript presents convincing data that clusters of cells active at specific locations during RUN were also activated together during REST2. However, after reading the manuscript, it is hard to have a clear idea what the REST2 synchronous activities looked like and whether they were truly offline reactivation as claimed.

i) It is unclear what behavioral state the cortex was in REST1 and REST2. The presence of hippocampal SWRs is commonly regarded as evidence of an offline state, although it could occur when the animal stays awake (awake ripples or replay). Since local field potentials (LFPs) were not recorded from M2, it is unknown whether the cortex was in a state (with LFP marks) different from the active behavior in RUN. Although the authors removed the time periods when animals were actively moving in REST1 and REST2, the cortex could be still active even during immobility.

The reviewer inquires whether the secondary motor cortex is in different states across the three behavioural epochs. Specifically, whether during the awake quiescent states, the cortex displays dynamics that distinguishes it from active running periods, and whether REST1 and REST2 differ in their states. As the reviewer pointed out, we did not acquire LFP from the cortex. We instead conducted analyses on the MUA to infer population dynamics.

Actions taken:

We conducted spectral analyses on the MUA across the three behavioural blocks. Briefly, we found that, during RUN, there was a marked increase in the ultra-slow oscillatory power (0.05-0.5 Hz). This increase appears to be due to an entrainment of population activities by locomotion. In contrast, slow oscillations power was increased during REST1 and REST2, which is consistent with a more synchronized cortical state typically observed during awake quiescence. Overall, these results suggest that the secondary motor cortex transitions to different cortical states between RUN and REST. However, no marked differences were observed in states between REST1 and REST2. Specific details of this analysis can be found in:

- Supplementary Fig. 1
- Results section (Lines 123-132)

ii) The manuscript does not quantify the duration of the identified cue/trajectory ensembles. The examples in Fig. 3a seem to have durations about a second. This is quite long compared to previously published cortical reactivation events. More importantly, how was the durations of ensembles in REST2 compared to those during RUN? Was there compression in the reactivation as previously shown?

The reviewer is concerned about the duration of the reactivation events and whether they are consistent with those reported in published literature. In the original manuscript, we quantified the durations of the reactivation events (Fig. 4g) and found them to be on the orders of ~100-200 ms, which are consistent with durations typically reported (e.g., Peyrache et al. 2009, Nat Neuro; Davidson, Kloosterman & Wilson 2009, Neuron; Kaefer et al. 2020, Neuron). In addition, the reviewer was interested in how these durations compared to RUN sequences (i.e., if there is temporal compression).

Actions taken:

We conducted temporal compression analysis to quantify the duration of reactivation events as compared to durations of RUN sequences. Briefly, we found that trajectory ensembles showed significantly more temporal compression compared to cue ensembles. This is consistent with the notion that cue ensemble neurons, as they are presumed to respond simultaneously to cue sensation, would also fire simultaneously during reactivation. In contrast, the trajectory sequences during RUN were estimated to be compressed ~30-fold during reactivation epochs. Details about this analysis and the results can be found in:

- Supplementary Fig. 9
- Results section (Lines 244-253)

3. Cortical-hippocampal-cortical loop

In the abstract and the discussion (Fig.6, Lines #577-583), the authors concluded that cue ensembles were reactivated in the cortex first. This cued the hippocampus to reactivate spatial representations, which in turn reactivated the cortical trajectory ensembles. I am puzzled how the conclusion was reached.

i) There are no data that support this temporal order of reactivation. If we assume hippocampal reactivation mostly occurred around SWRs, the presented data only show that both cue/trajectory reactivations in M2 occurred later than hippocampal ones (Fig. 3) and perhaps cue reactivation preceded trajectory reactivation (Fig. 4). But there are no data to show hippocampal SWRs were occurring between cue and trajectory reactivation events.

ii) The timing relationship plots in Figs. 3&4 show mostly broad peaks with long time delays (~hundreds milliseconds). It is understandable that the slow sampling rate (19 frames per second) limited the accuracy in determining temporal relations. Given that reactivation is commonly believed to be short events, it seems difficult to make the main conclusion from this type of data with limited temporal resolution.

The reviewer found the claim that cortical reactivations precede those of the hippocampus, which in turn is followed by cortical replay of trajectories, to be unsupported by the results presented. Indeed, as the reviewer pointed out, the slow sampling rate of calcium imaging makes it very hard to estimate precise timing relationships. This is further compounded by the fact that GCaMP imaging is an indirect measure of cellular activity, making it unfair to perform a direct comparison with the ephys data collected from CA1 and to estimate precise timings from it. In the original manuscript, we proposed the “cortical-hippocampal-cortical” loop and pattern completion models to help conceptualize the collection of our results within a comprehensive theoretical framework. We acknowledge that doing so would have conveyed the wrong message, leading to possible misinterpretations and confusion.

Actions taken:

- The abstract was rewritten, and the offending claims were removed (Lines 22-24)
- We re-emphasized that these are hypothetical models in the relevant Discussion sections (Lines 596, 627, 636-637) and Fig. 7
- Supplementary Fig. 12a-b were introduced to characterize the onset times of reactivations from the onsets of SWRs, on a per ensemble basis.

Reviewer #3 (Remarks to the Author):

Over the past several decades, each wave of results on memory trace reactivation has further underscored its importance and how much remains unknown. In this report, Chang and co-authors take a major step toward understanding the dynamics of reactivation by examining relationships between two different kinds of neural ensembles within premotor cortex. The study is important, the dataset impressive, and the analyses are convincing and relatively thorough. These results will be important not only within the neural systems field but more widely to anyone interested in how the brain processes and consolidates information. With respect to weaknesses, the authors unexpectedly omit analysis of the spatial relationship between representation types on the cortical surface, which is relevant to the stated justification of examining different dimensions in the “same” cortical region. Additionally, several minor edits would improve clarity and interpretation.

We thank the reviewer for highlighting the impact of our work and for their invaluable suggestions to help improve it. Please find below answers to your specific comments.

Major points

1. The one major concern relates to the absence of information on how neurons within and between ensembles are distributed across the cortical surface. This information would be valuable for, first, understanding how anatomically distributed the representations are (are ensembles with more trajectory information more anatomically localized, or dispersed, than the "cue" ensembles?) and, secondly, for confirming that the respective pools of multi-selective cells are, in fact, in the same cortical region. Line 122 states “[T]he current study aimed to explore whether these distinct representations, which are likely of hippocampal and cortical origin respectively, are reactivated concurrently by the same cortical region.” However, the size of the imaging window is large enough to sample multiple subdivision of premotor cortex, raising questions about the extent to which the ensembles are in fact in the same “region”.

The reviewer was concerned with the lack of topographic characterization in the present study, which could help determine whether cue and trajectory ensembles indeed coexist within the same cortical region or that they belong to anatomically separate subdivisions of the secondary motor cortex. In addition, this comment engendered some interesting avenues of exploration that could help further strengthen the main findings reported. Their comment can be elaborated into three specific research questions:

1. Do different subdivisions of the secondary motor cortex express different degrees of cue and trajectory bias?
2. Do cue ensemble and trajectory ensemble neurons show regional specificity? I.e., are they localized to specific subdivisions of the secondary motor cortex or are they anatomically overlapping?
3. Do cue and trajectory ensembles show different degrees of clusteredness/dispersion in topographic space?

Actions taken:

Topographic analysis was conducted to address these three questions. A new results section (Lines 501-539) and supplementary figure (Supplementary Fig. 18) were added to accommodate these new findings. Briefly, we found that:

1. Cue and spatial encoding lacks clear overall topographic organization in the secondary motor cortex. Nevertheless, a few patches with heightened densities of either cue or positional coding could be discerned. Therefore, neuronal coding may be locally organized.
2. Trajectory ensemble neurons appear to be uniformly distributed over the span of the imaging FOV. In contrast, cue ensemble neurons are more concentrated along the lateral aspect of M2. Therefore, although cue ensembles express some degree of topographic organization, there is a significant anatomical overlap between the two classes.
3. On the basis of individual ensembles, cue and trajectory ensembles exhibit the same degree of clusteredness in topographic space. However, the majority of ensembles tend to be clustered rather than dispersed, suggesting that their recruitment is not random, irrespective of ensemble class.

Minor points

Introduction

- Line 113: “Given that diverse cortical areas encompass a wide range of cognitive processes, the functional links that would permit associations to be formed between the hippocampus and the

cortex remains to be elucidated.” Here “given” seems to be the wrong word (Why does the former imply the latter?)

As the reviewer pointed out, this sentence lacks clarity and, upon reading it again, can in fact be altogether omitted without sacrifice to the semantics.

Actions taken:

The sentence was erased and a connector was added to the subsequent sentence: “Specifically, the spatial and non-spatial aspects of experiences, which seemingly constitute an important functional basis for the hippocampal-cortical dialogue (Buzsáki, 1996) in a distributed memory system, require further reconciliation.” (Lines 72-74)

Results

- Line 142 “the network, following locomotion, exhibited more stereotyped patterns of activity characterized by sparse population vectors”. There are a greater number of ensembles, with on average more neurons each, why are these necessarily “more sparse” as population vectors? Why would this be hypothesized? Sparsity can mean many different things, and immediately after this statement the vectors are referred to as temporally sparse. It would be useful to unpack in each case what the authors mean by sparsity.

The reviewer identified an inaccurate use of the terms "population sparsity" and "temporal sparsity". In the first instance, because the number of neurons per ensemble is comparable between REST1 and REST2, at individual reactivation/co-activation epochs, the population sparsity should be expected to be similar as well, as the reviewer implied. In the second instance, temporal sparsity was used to describe the lower rate of activation of REST2 ensembles. The use of this term here can generate confusion as temporal sparsity is usually used to describe the peri-stimulus responses of single neurons.

Actions taken:

The offending lines have been rephrased to more accurately reflect the actual finding, i.e., the network exhibits more stereotyped patterns of activity at the population level (Lines 137-146).

- Are differences between temporal sparsity during REST 1 and 2 meaningful, given the lack of apparent controls for sleep (or simply fatigue) and time over the rest intervals?

The reviewer hinted that the differences in temporal dynamics at the network level between REST1 and REST2 may be attributed to fatigue or other differences in behavioural states following active running. This is indeed a major limitation of the head-fixed preparation; mice do not fall asleep naturally unless trained under special conditions as some groups have shown. As such, differences in awake brain states are difficult to account for.

Actions taken:

- A passage was inserted into the Discussion to describe this limitation (Lines 580-584).
- We also included spectral analyses on the MUA to characterize the cortical states across the three behavioural epochs (Supplementary Fig. 1).

- Obviously there is no way to test whether REST 1 ensembles were also spatially selective for spaces that the mouse sampled before rest was recorded. It may be important to qualify that “spatial selectivity” refers specifically to selectivity of spaces in the behavioral apparatus. The

claim that the ensemble neurons were “drawn at random from the sample population” seems unwarranted, particularly without an analysis of the anatomical distribution of the ensemble cells in REST 1 versus REST 2.

The reviewer identified a claim that extrapolated beyond what the results/analysis actually conveyed, i.e., REST1 ensemble neurons were drawn at random from the total population. Indeed, the results from the aforementioned topographic analysis suggest that neurons of resting state ensembles tend to be more anatomically clustered, which suggest a systematic recruitment as opposed to a random process. Though the claim of randomness is of interest to forward replays (possibly from memory traces of previous days) or preplay (possible preconfigured map) phenomena, given that these are not the focus of the present article and that no adequate analyses were conducted to characterize them, such a claim is indeed unwarranted.

Actions taken:

The sentence was rephrased to more accurately reflect the actual finding. We also specified that "spatial selectivity" refers to the current session's RUN epoch: "This suggests that the constituent neurons of REST2 ensembles were more likely to include cells that were spatially selective during RUN than REST1 ensembles." (Lines 155-157)

- Line 228 "Both trajectory and cue ensembles contained a significantly higher number of spatially-selective cells than the ensembles that were unclassified." Wasn't spatial selectivity required for ensembles to be classified as cue or trajectory? If that's the case, then is this claim circular? (Ensembles were classified on account of spatial selectivity -> analysis showed that spatial selectivity was higher in classified ensembles)

The reviewer identified a substantial bias in the comparisons between fractions of spatially-selective cells contained within each ensemble category. Indeed, as per our selection criteria, only ensembles with 3 or more spatially-selective cells were qualified for either cue or trajectory ensemble classes. Therefore, it is possible that the significantly lower fractions reported for unclassified ensembles may be biased by our criteria.

Actions taken:

We re-conducted the analysis after discarding unclassified ensembles containing less than 3 spatially-selective cells for a fair comparison. In the original manuscript, there were 528 unclassified ensembles. Following removal, only 72 remained, suggesting that a bias was indeed present. Statistical comparisons yielded, however, the same results. Therefore, the conclusions drawn from this analysis remain valid. The figure and caption have been updated with the unbiased comparisons.

- Line 237 "Trajectory ensembles were more than three times as prevalent". How does this reflect the apparent difference in proportion of "cue" versus "trajectory" neurons? (Or, if difficult to classify as single neurons, the distribution of "cue-like" versus "trajectory like" using the model-fitting method.)

The reviewer was interested in a quantification of the proportions of cue-encoding and position-encoding neurons, at the single cells level. This could help offer a baseline comparison with the proportions reported for cue vs. trajectory ensembles. Ideally, this question would have been tested experimentally by including sessions with a blank belt without any visuo-tactile cues and identifying neurons that become active with the introduction of the cues. Given that was not the case in the current study, we could only offer a rough estimation by inference.

Actions taken:

A method was devised to obtain an estimate of the proportion of cue-encoding neurons from the total population spatially-selective cells. The following were added:

- Results (Lines 230-232)
- Supplementary Fig. 7

- In the section “Cue and trajectory ensembles reactivate for complementary features” it is not immediately obvious that pairs of “cue and trajectory” refers specifically to cue-trajectory pairings. On first read, this appeared to mean “pairs of cue ensembles, pairs of trajectory ensembles, and pairs of cue-trajectory ensembles”. Some change in the language (e.g., “cue-trajectory pairs”) would be useful to make this section easier to follow on an initial read.

Actions taken:

The section has been rephrased for clarity. Specifically, we used the wordings “cue-trajectory ensemble pairs” as suggested by the reviewer, to make the section easier to follow.

Discussion

- The difference in day-to-day stability between cue and trajectory ensembles is very interesting, as it potentially offers a glimpse into how space/trajectory information is encoded and consolidated in M2 over learning. To put another way, the data hint at how contextual information may influence M2 activity specifically for some dimensions and not others. An alternative explanation for this result might be that the “partial remapping” of trajectory information over days relates to minor changes in movement behavior between cue locations, compared with "at" cue locations. Importantly, these two interpretations aren't exclusive, and in fact could be related. Either way, it may be nice to see more elaboration on this result in the Discussion, ideally with some reference to work by Doug Nitz and colleagues illustrating how M2 may “transform spatial information into planned action during navigation” (Olson et al., 2020).

We share much of the same enthusiasm as the reviewer about these results. Indeed, the interactions between spatial and non-spatial information in the cortex could hypothetically subserve a mechanism for both online memory-guided behaviours/navigation, as well as offline consolidation for modifying existing cortical "schemas/engrams". An extensive body of works by Nitz, Knierim, Wilber and colleagues have identified areas, such as the retrosplenial cortex and the posterior parietal cortex, involved in transcoding allocentric spatial representations into egocentric features, which can be subsequently used for the planning of actions. The study referenced by the reviewer further extended this property to the M2, where contextual information appears to integrate with action plans, similar to the conjunctive encoding of spatial and non-spatial information shown in the present study. As a future direction, it would be interesting to study whether the integration of spatial information into cortical representations is a gradual process that systematically drives representational drifts with experience and learning (cf. Kira et al., 2023, Nat Comm).

Actions taken:

We elaborated on these possibilities in the Discussion section (Lines 636-660).

REVIEWERS' COMMENTS

Reviewer #1 (Remarks to the Author):

The authors have done an admirable job addressing my comments. The text additions and edits to the Discussion/Results provide additional context and helpful interpretation. Also, the new metrics presented in the Supplemental Material further strengthen the core claims of the study. This is a solid study that advances our understanding of the mechanisms of offline memory consolidation.

Reviewer #2 (Remarks to the Author):

The authors have adequately addressed the reviewers' concerns. I fully support the publication of this paper.

Reviewer #3 (Remarks to the Author):

The authors have adequately addressed all reviewer comments. As previously stated, the report provides a strong and detailed account of offline activity patterns for different types of information in premotor cortex. This advances our knowledge of offline reactivation as well as the transformation and consolidation of information in the premotor cortex.

The primary question I had after reading the previous version was whether we could truly consider the cue and trajectory information to be found in the "same region". If not, this would raise further questions about whether differential dynamics of cue- and trajectory-ensembles may be due to differences in circuitry or long-range connectivity between subregions. The authors now address the issue of topographical clustering of representation types in Supplementary Figure 18. Panel d of this figure in particular helps emphasize the intermixing of the information. While it may have been nice to see a distribution/histogram of Euclidean distances between the more cue-coding and more trajectory-coding neurons, the data as they stand already confirm that ensembles supporting both domains of information are highly overlapping topologically. They also offer an intriguing glimpse into clustering of information types, raising new questions that can be addressed in future work.

REVIEWERS' COMMENTS

Reviewer #1 (Remarks to the Author):

The authors have done an admirable job addressing my comments. The text additions and edits to the Discussion/Results provide additional context and helpful interpretation. Also, the new metrics presented in the Supplemental Material further strengthen the core claims of the study. This is a solid study that advances our understanding of the mechanisms of offline memory consolidation.

We thank the reviewer for their constructive comments throughout this review process, which we believe have helped to improve the manuscript substantially.

Reviewer #2 (Remarks to the Author):

The authors have adequately addressed the reviewers' concerns. I fully support the publication of this paper.

We thank the reviewer for their meticulous examination of our manuscript, and we believe that their comments have helped strengthen the core claims of our study considerably.

Reviewer #3 (Remarks to the Author):

The authors have adequately addressed all reviewer comments. As previously stated, the report provides a strong and detailed account of offline activity patterns for different types of information in premotor cortex. This advances our knowledge of offline reactivation as well as the transformation and consolidation of information in the premotor cortex.

The primary question I had after reading the previous version was whether we could truly consider the cue and trajectory information to be found in the "same region". If not, this would raise further questions about whether differential dynamics of cue- and trajectory-ensembles may be due to differences in circuitry or long-range connectivity between subregions. The authors now address the issue of topographical clustering of representation types in Supplementary Figure 18. Panel d of this figure in particular helps emphasize the intermixing of the information. While it may have been nice to see a distribution/histogram of Euclidean distances between the more cue-coding and more trajectory-coding neurons, the data as they stand already confirm that ensembles supporting both domains of information are highly overlapping topologically. They also offer an intriguing glimpse into clustering of information types, raising new questions that can be addressed in future work.

We thank the reviewer for their insightful comments and would like to remark that it has been a true pleasure to engage in these curiosity-driven discussions with them during this review process. In particular, we believe that the topographic analyses suggested by the reviewer was a nice addition to make this study more complete.